



# The surface tension and CCN activation of sea spray aerosol particles

Judith Kleinheins[1], Nadia Shardt[2], Ulrike Lohmann[1], and Claudia Marcolli[1]

[1]Institute for Atmospheric and Climate Science, ETH Zürich, Universitätsstrasse 16, 8092 Zürich, Switzerland
[2]Department of Chemical Engineering, Norwegian University of Science and Technology (NTNU), 7491 Trondheim, Norway

**Correspondence:** Judith Kleinheins (judith.kleinheins@env.ethz.ch)

**Abstract.** In marine environments, sea spray aerosol (SSA) particles have been found to contain surface-active substances (surfactants) originating from the sea surface microlayer. These surfactants can lower the surface tension of the SSA particles, facilitating their activation to cloud droplets. This effect is not considered in classical Köhler theory, which assumes droplets to be homogeneous with a surface tension of pure water. In this study the CCN activity of SSA particles calculated with classical Köhler theory is compared to a more complex calculation that considers bulk–surface partitioning, surface tension lowering, and liquid–liquid phase separation. The model approach presented here combines the multi-component Eberhart model for surface tension with the Monolayer model and an activity model (AIOMFAC). This combination allows for the first time to calculate Köhler curves of surfactant-containing particles with a large number of compounds. In a sensitivity study we show that organic compounds can be categorized into weak, intermediate, and strong surfactants for CCN activation based on their separation factor in water $S_{1i}$ and their pure component surface tension $\sigma_i$. For a quaternary model system of SSA particles, it is shown that a high content of hydrophobic organic material (i.e., strong surfactants) in Aitken mode particles does not necessarily prevent good CCN activation, but rather facilitates effective activation via surface tension lowering. Since common climate models use parametrizations that are based on classical Köhler theory, these results suggest that the CCN activity of small SSA particles might be underestimated in climate models.

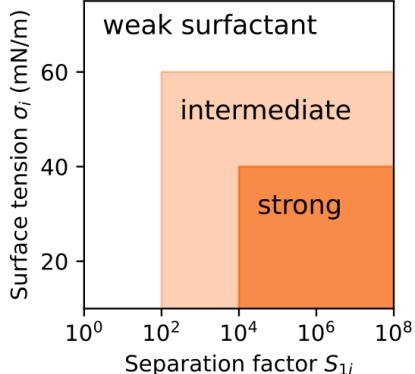

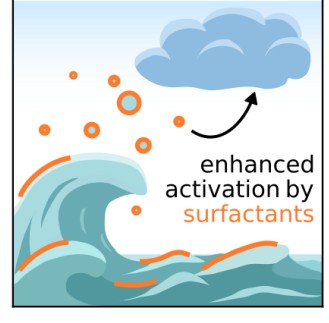



## 1 Introduction

The oceans are covered to a large extent by an organic-rich layer, the so-called sea surface microlayer (SSM, Wurl et al., 2011). Since about 70% of the Earth's surface is covered by oceans, the SSM constitutes a large part of the interface between the condensed mass of the planet and its gaseous atmosphere. Its coverage is likely to even increase as climate change progresses and sea ice melts (Christiansen et al., 2020). The SSM contains surface-active organic substances such as lipids, fatty acids, polysaccharides, and proteins, as well as biota (e.g., bacteria and fish larvae), and substances of anthropogenic origin (e.g., plastics and tar lumps, Hardy, 1982; Wurl and Holmes, 2008). Aerosolization of the SSM by bubble bursting and wave breaking leads to sea spray aerosol (SSA) particles with a high organic content (Facchini et al., 2008; O'Dowd et al., 2004). The presence of surface-active substances (surfactants) in aerosol particles can result in a surface tension lower than that of pure water or salt solutions, which reduces the Kelvin effect in the Köhler equation (Köhler, 1936). This raises the question whether surface-active material facilitates cloud formation by enhancing cloud condensation nucleus (CCN) activation.

Classical Köhler theory, which is used as a basis for state-of-the-art climate modelling, does not take surface tension effects into account but assumes a surface tension of pure water for all aerosol particles. In order to analyze the potential error introduced by this assumption, a number of studies have investigated the potential influence that bulk–surface partitioning and surface tension lowering could have on CCN activation, yet no general agreement has been reached. While some studies suggest an enhanced CCN activation even for particles containing weakly surface-active substances (e.g., small dicarboxylic acids, Ruehl et al., 2016; Sareen et al., 2013), other studies find an effect only for strongly surface-active substances or droplets with high organic content (Ovadnevaite et al., 2017; Frosch et al., 2011; Kokkola et al., 2006; Lohmann and Leck, 2005), and even other studies find no strong enhancement of CCN activity or cloud droplet number concentrations at all (Forestieri et al., 2018; Kristensen et al., 2014; Prisle et al., 2010; Lohmann et al., 2004).

The reason why these studies report seemingly contradictory results is that they greatly differ in the systems that were analyzed (aerosol particle size and composition) as well as in their experimental methods or modelling approaches. On the experimental side, difficulties exist in determining the molecular composition of field samples. For laboratory-generated aerosol particles, challenges arise in producing particles with an accurate surfactant content as well as minimizing impurities. On the modelling side, challenges include (i) modelling the surface tension of complex mixtures including non-ideality effects like salting-out (El Haber et al., 2023), (ii) accounting for bulk depletion caused by the high surface-to-volume ratio in small droplets (Vepsäläinen et al., 2022, 2023), and (iii) determining the water activity of solutions containing surfactants. As a result, the relevance of surface effects on cloud formation is still uncertain despite considerable research efforts.

One of the first studies considering bulk depletion due to bulk–surface partitioning was that by Sorjamaa et al. (2004) using a model based on Gibbsian thermodynamics ("Gibbs model"). In the Gibbs model, the surface is described as an infinitely thin layer between the bulk phase and the gas phase—the so-called Gibbs dividing surface—and the Gibbs adsorption equation is used to relate the surface composition to the surface tension. Another partitioning model was suggested by Malila and Prisle (2018) based on the assumption that the surface phase consists of a one-molecule-thick layer ("Monolayer model"). In this model, the surface tension is given by the average of the pure component surface tensions weighted by the substances' surface





volume fractions. Besides these two models, various other approaches to model the bulk–surface partitioning in small droplets have been suggested in the past. In Vepsäläinen et al. (2022, 2023), six common model approaches are compared. For soluble organic acids (Vepsäläinen et al., 2022), it is concluded that the "Gibbs model" (Prisle et al., 2010) and the "Monolayer model" (Malila and Prisle, 2018) are the current preferable options for modeling droplet growth and activation. For strongly surface-active substances (Vepsäläinen et al., 2023), no clear conclusion is presented as to which model can be recommended. Instead, it is shown that the models yield largely different surface tensions and critical supersaturations. Recently, Bzdek et al. (2020) and Bain et al. (2023) found good agreement when they compared the Monolayer model to experimental results obtained from micron- and submicron-sized particles, suggesting the validity of this approach.

In this study, we address the challenges related to modelling surfactant-containing aerosol particles with a bottom-up validated modelling approach. The model we present is able to take into account (i) non-ideality effects in surface tension, (ii) bulk depletion, and (iii) non-ideality in the bulk phase. Based on this model approach, we aim to provide an overview of which particle sizes and compositions necessitate the inclusion of surface effects for the calculation of the critical supersaturation and for what systems classical Köhler theory suffices. We focus on SSA due to its high abundance and since it is well known to contain strong surfactants (Bertram et al., 2018).

## 2 Modelling approach

The vapor pressure of a solution droplet was derived by Köhler (1936) as

$$SS = \left( a_w \exp \left( \frac{4\sigma v_w}{D_{\mathrm{wet}} RT} \right) - 1 \right) \cdot 100\%, \tag{1}$$

with the supersaturation $SS$ in percent, the water activity $a_w$, the surface tension $\sigma$, the molar volume of water $v_w$, the wet diameter $D_{\mathrm{wet}}$, the universal gas constant $R = 8.314 \, \mathrm{J \, m^{-1} \, K^{-1}}$, and the temperature $T$. In this work, calculations are performed at room temperature, i.e., $T = 25\,°\mathrm{C}$ unless stated otherwise. The critical supersaturation $SS_{\mathrm{crit}}$ is given by the maximum of $SS$ over $D_{\mathrm{wet}}$. Since focus is laid on modelling the CCN activation, which requires supersaturated conditions, the particles are assumed to be in liquid state at all considered levels of humidity.

In classical Köhler theory, solution ideality and the surface tension of water $\sigma_1$ are assumed. While these assumptions are justified for purely inorganic particles, the presence of organic substances can lead to non-ideal mixing and a lowering of the surface tension compared to that of water. To quantify the error that is introduced by assuming classical Köhler theory for surfactant-containing particles, we calculate $SS_{\mathrm{crit}}$ for SSA particles for both classical Köhler theory and a more complex model approach which accounts for solution non-ideality and surface tension lowering, as described in the following.

### 2.1 Composition-dependent surface tension

An atmospheric aerosol particle containing surfactants has a lower surface tension than a particle without surfactants, and the extent of surface tension lowering depends on how dilute the particle is. At low relative humidity, a surfactant-containing particle in a deliquesced state is expected to have a low surface tension. However, with increasing humidity, the particle dilutes





and as a result, the surface tension increases and approaches the value of pure water, as illustrated in detail by Davies et al. (2019). Therefore, a surface tension model is required to quantify surface tension as a function of the solution composition.

For aqueous mixtures with one solute (binary solution), numerous surface tension models have been proposed in the past. A number of binary surface tension models were reviewed in Kleinheins et al. (2023) and tested for a broad range of compounds, showing that the experimental surface tension data at a fixed temperature (surface tension isotherm) closely follows a sigmoidal curve when plotted on an x-axis with a logarithmic mole fraction. Based on the logistic function, the Sigmoid model was derived as:

$$\sigma = \sigma_1 - (\sigma_1 - \sigma_i)\left(10^{pd} + 1\right)\frac{x_i^d}{10^{pd} + x_i^d}, \tag{2}$$

where $x_i$ is the mole fraction of the solute $i$ and $\sigma_1$ and $\sigma_i$ are the pure component surface tensions of water and a solute $i$, respectively. The parameters $p$ and $d$ can be obtained by fitting Eq. 2 to experimental data and they characterize the inflection point ($p$) and the distance ($d$) of the inflection point from the critical micelle concentration (CMC). According to Kleinheins et al. (2023), with $p$, $d$, and $\sigma_i$, the surface tension isotherm of a substance is sufficiently characterized.

Among these parameters, $d$ is of minor importance. El Haber et al. (2024) report binary aqueous surface tension data of over 130 organic compounds and provide fits with the Sigmoid model for 56 of the compounds. From all fits, $d$ has an average value of 1.3, and 80 % of all fits result in $d$ between 0.6 and 1.7. In a simple sensitivity calculation, a variation of $d$ in this range was found to have only a small influence on $SS_{\mathrm{crit}}$, yielding a maximum deviation of $\Delta SS_{\mathrm{crit}} \approx 0.05$ % under extreme conditions (see supplement Sect. S1).

When choosing $d = 1$, the Sigmoid model simplifies to a function that is mathematically equivalent to the Eberhart model (Eberhart, 1966). Instead of a parameter $p$, the Eberhart model uses a separation factor $S$, that describes the adsorption–desorption rate of the solute relative to the solvent. Using a subscript 1 for water and $i$ for the solute, the model is written as

$$\sigma = \frac{\sigma_1(1 - x_i) + \sigma_i S_{1i} x_i}{(1 - x_i) + S_{1i} x_i}. \tag{3}$$

Mathematically, $S_{1i}$ is related to $p$ via $10^p = 1/(S_{1i} - 1)$. If $S_{1i} = 1$, water and the solute $i$ have the same adsorption–desorption rate to the surface such that no substance is enriched at the surface. If $S_{1i} > 1$, the solute $i$ has a preference to partition to the surface while if $S_{1i} < 1$, the solute $i$ is expected to be depleted at the surface. Following the Eberhart model, we can fully characterize the surface activity of substances in a binary solution with water with only two parameters, i.e., $S_{1i}$ and $\sigma_i$.

Atmospheric aerosol particles are complex mixtures with more than one solute. Therefore, to accurately calculate their surface tension, a multi-component surface tension model is required. Based on the binary Eberhart model and the multi-component Connors-Wright model by Shardt et al. (2021), a multi-component Eberhart model was derived by Kleinheins et al. (2024) as:

$$\sigma = \sum_{i=1}^{n} \sigma_i x_i + \sum_{i=1}^{n}\left(\frac{x_i}{\sum_{j=1}^{n} x_j/S_{ji}} \sum_{j=1}^{n} \frac{x_j}{S_{ji}}(\sigma_j - \sigma_i)\right), \tag{4}$$





where $S_{ij}$ is the separation factor of a binary mixture of substances $i$ and $j$. This model was found to predict accurately the surface tension of ternary ideal mixtures if all $S_{ij}$ values are known from binary surface tension data. If a factor $S_{ij}$ is unknown, it can be obtained by fitting it to ternary surface tension data.

Ternary solutions containing surfactants in a mixture with a salt were found to exhibit salting-out effects leading to an in-
creased surface concentration of the surfactant (Kleinheins et al., 2024; El Haber et al., 2023). This non-ideal surface tension behaviour can be taken into account by additional parameters $A_{ij}^{\mathrm{SO}}$ and $B_{ij}^{\mathrm{SO}}$, for surface and bulk related non-ideality, respectively. When considering surface related non-ideality ($A_{ij}^{\mathrm{SO}} \neq 0$), the surface tension of the surfactant $\sigma_i$ is perturbed as a function of the mole fraction of the salt $x_j$ as

$$\sigma_i^{\mathrm{non-ideal}} = \sigma_i \left(1 - x_j A_{ij}^{\mathrm{SO}}\right). \tag{5}$$

For bulk-related non-ideality, the separation factor of the surfactant in water $S_{1i}$ is perturbed as

$$S_{1i}^{\mathrm{non-ideal}} = S_{1i} \left(1 + x_j B_{ij}^{\mathrm{SO}}\right). \tag{6}$$

The salting-out factors $A_{ij}^{\mathrm{SO}}$ and $B_{ij}^{\mathrm{SO}}$ can be obtained by fitting the model (Eq. 4– 6) to ternary surface tension data (Kleinheins et al., 2024).

## 2.2  Bulk–surface partitioning and bulk depletion

In large liquid volumes, the bulk composition $x_i^{\mathrm{bulk}}$ can be assumed to be equal to the total composition $x_i^{\mathrm{tot}}$ (i.e., $x_i = x_i^{\mathrm{bulk}} \approx x_i^{\mathrm{tot}}$) in surface tension isotherms (e.g., Eberhart model). Small droplets, however, have a large surface-to-volume ratio and therefore the partitioning of substances to the surface of the droplet can lead to their depletion in the droplet bulk. To take this effect into account, a partitioning model is required that introduces mass conservation and allows to quantify the bulk depletion based on physical and geometrical assumptions. Among the partitioning models described in literature, the Monolayer model
(Malila and Prisle, 2018) is the only one that has been validated with microscopic surface tension data (Bzdek et al., 2020; Bain et al., 2023). Therefore, in this study, this model is chosen for bulk–surface partitioning.

Briefly, the monolayer model calculates the surface tension of a particle based on its surface composition and an experimentally constrained surface tension isotherm. To do so, the droplet is divided into a spherical bulk volume $V^{\mathrm{bulk}}$ and a surface volume $V^{\mathrm{surf}}$, which has the shape of a spherical shell with a thickness of one molecular layer. The composition of the bulk and surface phases ($x_i^{\mathrm{bulk}}$ and $x_i^{\mathrm{surf}}$) and the surface tension $\sigma$ of the droplet are determined using the total composition $x_i^{\mathrm{tot}}$
and the wet diameter $D_{\mathrm{wet}}$ of the droplet as input parameters. As a first constraint, mass conservation has to be fulfilled, such that

$$n_i^{\mathrm{tot}} = n_i^{\mathrm{bulk}} + n_i^{\mathrm{surf}}, \tag{7}$$

where $n_i$ is the number of molecules of substance $i$ in the entire droplet (tot), the surface phase (surf), or the bulk phase
(bulk). Note that $x_i^{\mathrm{tot}} \neq x_i^{\mathrm{bulk}} + x_i^{\mathrm{surf}}$. In the model suggested by Malila and Prisle (2018), composition-dependent density parametrizations are used. In contrast, for simplicity we assume volume additivity in all phases as $V = \sum_i n_i v_i$, where $v_i$ is





the molecular volume of substance $i$, which is calculated as $v_i = M_i/(\rho_i N_A)$ from the substance's molar mass $M_i$, its density $\rho_i$ and Avogadro's number $N_A$. With these relationships, the number of molecules in each phase can be calculated from the volume and the composition of the phase.

For the calculation of $V^{\mathrm{surf}}$, the monolayer thickness $\delta$ is required, which is assumed to depend on the monolayer composition as

$$\delta = \left( \frac{6}{\pi} \sum_i v_i x_i^{\mathrm{surf}} \right)^{1/3}. \tag{8}$$

Thus, $\delta$ is a number-weighted average of the equivalent diameter of the molecules in the surface phase, where the equivalent diameter of a molecule is given by the diameter of a sphere with the same volume as the molecular volume $v_i$.

In addition to these geometrical assumptions, the model requires information on (i) the strength of partitioning of a substance to the surface and (ii) how the surface composition relates to surface tension. The strength of partitioning of a substance is derived from a surface tension isotherm

$$\sigma = f(x_i^{\mathrm{bulk}}) \tag{9}$$

which can be obtained from macroscopic surface tension measurements, where $x_i^{\mathrm{bulk}} = x_i^{\mathrm{tot}}$. Here, we use the multi-component

Eberhart model (Kleinheins et al., 2024) in contrast to Malila and Prisle (2018), who used a Szyszkowski–Langmuir based equation, which is limited to ternary solutions. The surface tension $\sigma$ is related to the surface composition via

$$\sigma = \frac{\sum_i \sigma_i v_i x_i^{\mathrm{surf}}}{\sum_i v_i x_i^{\mathrm{surf}}}, \tag{10}$$

i.e., the surface tension is an average of the pure component surface tensions, weighted by the surface volume fractions, which is a key assumption of the Monolayer model. Equations 7–10 provide a system of equations that can be solved iteratively for an

infinite number of substances with a nested pseudo-binary approach. The procedure used in this study to solve the monolayer model for a quaternary system is shown in more detail in supplement Sect. S2.

     Monolayer model calculations can also be solved taking surface tension non-ideality, e.g., salting-out, into account but doing so requires an adaptation in the Monolayer model equations. The enhanced partitioning and lowered surface tension due to salting-out can be introduced via the surface tension isotherm, i.e., the multi-component Eberhart model using Eq. 5 and 6.

If $A_{ij}^{\mathrm{SO}} > 0$, the surface tension of the mixture $\sigma$ can be lower than any of the pure component surface tensions ($\sigma < \min(\sigma_i)$). This is problematic when solving for the surface composition via Eq. 10. For consistency between the non-ideal Eberhart model and the Monolayer model, Eq. 10 needs to be replaced with

$$\sigma = \frac{\sum_i \sigma_i^{\mathrm{non-ideal}} v_i x_i^{\mathrm{surf}}}{\sum_i v_i x_i^{\mathrm{surf}}}. \tag{11}$$

### 2.3   Water activity and solution non-ideality

Once $x_i^{\mathrm{surf}}$, $x_i^{\mathrm{bulk}}$, and $\sigma$ are calculated with the Monolayer model, the water activity $a_w$ needs to be determined to finally calculate the critical supersaturation via Eq. 1. In classical Köhler theory, no bulk–surface partitioning is considered and



solution ideality is assumed, i.e., $a_w = \hat{x}_w^{\mathrm{tot}}$, where $\hat{x}_w^{\mathrm{tot}}$ is calculated based on $x_w^{\mathrm{tot}}$ and the dissociation of inorganic salts into ions (e.g., NaCl into 2 ions) and no dissociation of the organic substances. These assumptions are made in many studies. However, organic substances with a low oxygen-to-carbon (O:C) ratio display non-ideal mixing behaviour in water. Therefore,

for our best estimate of $SS_{\mathrm{crit}}$, we add a calculation of $a_w$ based on $x_w^{\mathrm{bulk}}$ considering solution non-ideality.

To do so, an activity coefficient $\gamma_w$ is introduced as $a_w = \gamma_w x_w^{\mathrm{bulk}}$. The group contribution model AIOMFAC (Zuend et al., 2008, 2011) allows to calculate activity coefficients for organic–inorganic mixtures. Here, we use the web version (https://aiomfac.lab.mcgill.ca) of AIOMFAC for the calculation of $\gamma_w$ and $a_w$. Note that sodium dodecyl sulfate (SDS)—a substance that is used in this study—cannot be represented with the functional groups implemented in AIOMFAC, due to the organic-

sulfate group. For the calculation of $a_w$ we therefore represent it with dodecanoic acid, which is a fatty acid with the same hydrocarbon chain length.

In the calculations with the web version of AIOMFAC, all substances are assumed to be in a single homogeneous phase and we label this calculation therefore "AIOMFAC-1ph". When substances with a low oxygen-to-carbon (O:C) ratio are mixed with inorganic salts in the same phase, this can lead to unphysically high activity coefficients being predicted by AIOMFAC.

In this case, the solution undergoes liquid–liquid phase separation (LLPS) (Zuend and Seinfeld, 2012; Ciobanu et al., 2009). When the organic substances are surfactants, micelles form at concentrations exceeding the CMC, which can be considered as a type of phase separation. In both cases, the organic substance is strongly depleted in the aqueous phase and its contribution to the Raoult effect is diminished. To consider this effect, we calculate $a_w$ assuming that the surfactant is totally hydrophobic and entirely present in a separate phase. In this case, the bulk mole fractions are converted to "surfactant-free" mole fractions

$\tilde{x}_i^{\mathrm{bulk}}$ to calculate $a_w$ with AIOMFAC. This approach gives the maximum limit of $a_w$ at small wet diameters when micelles and/or LLPS are present. To obtain a best estimate of $a_w$, we combine the AIOMFAC-1ph result at large wet diameters with the hydrophobic approach at small wet diameters by always taking the lower value of the two. More details are given in supplement Sect. S3.

## 3   Representation of sea spray aerosol particles

The marine aerosol can be divided into a primary aerosol produced by breaking waves (SSA) and a secondary aerosol produced by gas-to-particle conversion of volatile species (secondary marine aerosol) (Rinaldi et al., 2010). In this study, we focus on freshly emitted (nascent) SSA particles to investigate the influence of their high surfactant content on CCN activation. Note that atmospheric aging of SSA leads to fragmentation reactions of organics with OH resulting in organic mass loss (Trueblood et al., 2019) and uptake of sulfuric and nitric acid, methanesulfonic acid, and organic acids, resulting in chloride depletion

(Su et al., 2022). These aging processes are expected to increase the total fraction of soluble species and to potentially lower the fraction of large surface-active molecules. Therefore, by focusing on nascent SSA in this study, an upper estimate of the influence of surface effects on CCN activation is given.

In this study, we represent SSA by a quaternary model system as illustrated in Fig. 1 (A). Besides water (1), the SSA is assumed to contain surfactants (2), water-soluble organic compounds (WSOC) (3), and inorganic salts (4). For each of these





categories, one model compound is chosen to represent the group of substances. For the inorganic salts, NaCl is chosen as the model compound, because the inorganic fraction of nascent SSA was found to consist mostly of NaCl with only small amounts of calcium, magnesium and potassium chlorides, sulfates and carbonates (Bertram et al., 2018). The mass fractions of the organic substances and their model compounds are discussed in the following sections. An illustrative depiction of bulk–surface partitioning in a droplet with a given total composition is shown in Fig. 1 (C).

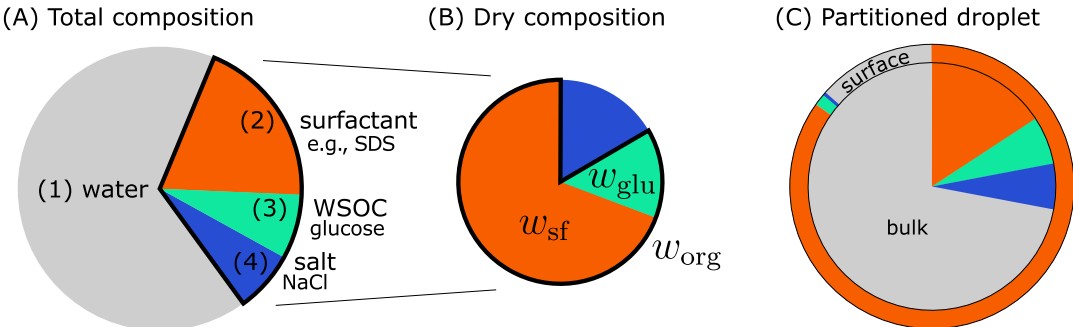

**Figure 1.** Schematic representation of the quaternary model SSA particle: (A) Total composition with model compounds; (B) Dry composition with organic fraction $w_{\mathrm{org}}$, glucose fraction $w_{\mathrm{glu}}$, and surfactant fraction $w_{\mathrm{sf}}$; (C) droplet with bulk–surface partitioning, where the surface is considered to comprise one molecular layer. Dimensions are not to scale.

### 3.1 Size dependent organic content

The chemical composition of SSA has been reviewed thoroughly by Bertram et al. (2018) and we follow their recommendations in this study. Besides inorganic salts, SSA contains organic compounds originating from the SSM. It was found that the organic-to-inorganic ratio depends strongly on the size of the particles, with smaller particles having a higher organic content. Figure 2 summarizes measurements of the mass ratio of organic carbon to sodium ions $\mathrm{mC/mNa^+}$ for a range of dry particle diameters, evidencing the increase in organic fraction with decreasing particle size (Bertram et al., 2018). To derive a simple relationship of the organic content in SSA particles as a function of size, a linear fit was made to the data resulting in

$$\log_{10}\left(\mathrm{mC/mNa^+}\right) = -0.448 - 1.37\log_{10}\left(D_{\mathrm{dry}}/\mu\mathrm{m}\right), \tag{12}$$

where $D_{\mathrm{dry}}$ is the dry diameter in micrometers. Note that in real SSA particles, the presence of $\mathrm{MgCl_2}$ and $\mathrm{CaCl_2}$ may lead to the formation of hydrates upon efflorescence (Rasmussen et al., 2017). Calculation of the dry salt mass based on a measured dry diameter of such a particle can be erroneous if these hydrates are not accounted for correctly. In this study, the dry diameter refers to the equivalent diameter of a totally dry particle, i.e., containing no hydrates.

Besides experimental uncertainty, natural variability in the organic content contributes to some spread of the data around the fitted mean. To take this spread into account, 99 % confidence intervals of the linear fit were computed providing a lower (case "low") and higher (case "high") limit of the organic content in addition to the average value given by the linear fit (case




"med"). Since no data is available for particles below 100 nm, the organic content of these particles is uncertain and can lie outside of the range covered by the cases "low"–"high". Therefore, in Sect. 4.2, the organic content is varied in a broader range for sensitivity analysis.

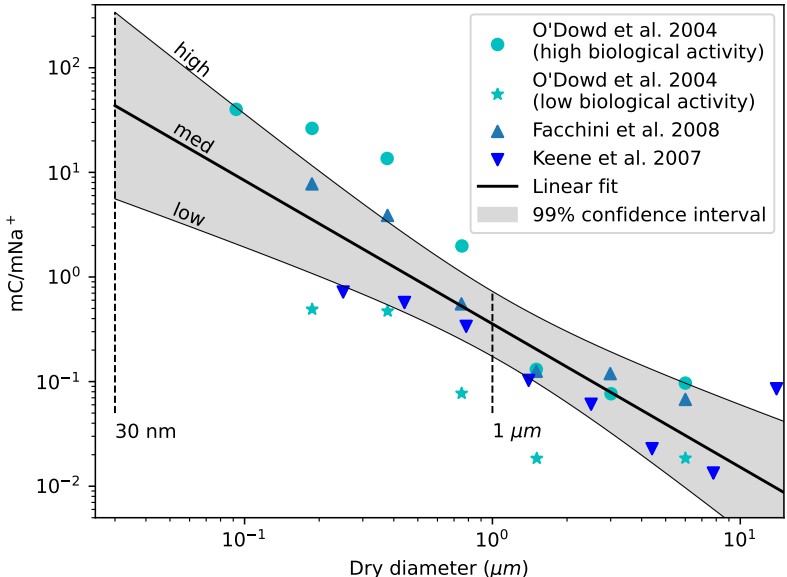

**Figure 2.** Mass ratio of organic carbon to sodium ions $\mathrm{mC/mNa^+}$ of SSA particles as a function of their dry diameter. Symbols: measurements from field (O'Dowd et al., 2004) and laboratory studies (Facchini et al., 2008; Keene et al., 2007) reproduced from Bertram et al. (2018). Solid lines: linear fit (Eq. 12, case "med") and 99 % confidence interval (cases "low" and "high").

We converted $\mathrm{mC/mNa^+}$ to an organic dry mass fraction $w_{\mathrm{org}}$ by using O:C and H:C elemental ratios of 0.34 and 1.43, respectively, following the suggestion from Bertram et al. (2018) for sub-micrometer particles. The resulting $w_{\mathrm{org}}$ for the three
cases "low", "med", and "high" and dry diameters from 30 nm to 1 µm are shown in Table 1.

**Table 1.** Organic dry mass fraction $w_{\mathrm{org}}$ for a range of dry diameters $D_{\mathrm{dry}}$ (nm) derived from the linear fit in Fig. 2 (case "med") and its 99 % confidence interval (cases "low" and "high").

| $D_{\mathrm{dry}}$ | 30 | 40 | 50 | 60 | 70 | 80 | 90 | 100 | 110 | 120 | 130 | 140 | 150 |
|---|---|---|---|---|---|---|---|---|---|---|---|---|---|
| high | 0.995 | 0.992 | 0.988 | 0.983 | 0.977 | 0.971 | 0.965 | 0.957 | 0.95 | 0.941 | 0.933 | 0.924 | 0.914 |
| med | 0.964 | 0.948 | 0.93 | 0.912 | 0.894 | 0.875 | 0.857 | 0.838 | 0.819 | 0.801 | 0.783 | 0.765 | 0.748 |
| low | 0.776 | 0.729 | 0.689 | 0.653 | 0.621 | 0.593 | 0.567 | 0.544 | 0.522 | 0.503 | 0.484 | 0.467 | 0.451 |
| $D_{\mathrm{dry}}$ | 200 | 250 | 300 | 350 | 400 | 450 | 500 | 600 | 700 | 800 | 900 | 1000 | |
| high | 0.864 | 0.811 | 0.756 | 0.704 | 0.655 | 0.609 | 0.566 | 0.493 | 0.433 | 0.384 | 0.344 | 0.311 | |
| med | 0.667 | 0.596 | 0.534 | 0.482 | 0.436 | 0.397 | 0.363 | 0.307 | 0.264 | 0.23 | 0.203 | 0.181 | |
| low | 0.386 | 0.336 | 0.298 | 0.266 | 0.24 | 0.218 | 0.199 | 0.168 | 0.145 | 0.126 | 0.11 | 0.097 | |





## 3.2 Composition of the organic fraction

The organic dry mass fraction given in Table 1 is subdivided into WSOC and surfactants. A large fraction of WSOC were identified as saccharides with glucose detected as the most abundant species (Bertram et al., 2018; Hasenecz et al., 2020; Jayarathne et al., 2016). Therefore, we represent WSOC with glucose and we define

$$w_{\mathrm{org}} = w_{\mathrm{glu}} + w_{\mathrm{sf}}, \tag{13}$$

where $w_{\mathrm{glu}}$ and $w_{\mathrm{sf}}$ are the dry mass fractions of glucose and the surfactant, respectively (see Fig. 1 (B)). The percentage of WSOC in the total organic mass was found to be very small ($\approx 5\,\%$ for particles with $D_{\mathrm{dry}} = 125 - 250\,\mathrm{nm}$, Facchini et al., 2008). Hence, as our best estimate, we set the fraction of glucose in the total organic mass $w_{\mathrm{glu}}/w_{\mathrm{org}} = 0.05$. In addition, $w_{\mathrm{glu}}/w_{\mathrm{org}}$ is also varied from 0 to 1 for sensitivity analysis in Sect. 4.

The molecular identity of surfactants in SSA has been analyzed by Cochran et al. (2016) with mass spectrometry. Their results suggest that the surfactants are mostly composed of saturated fatty acids, with palmitic (hexadecanoic) acid as the most abundant species. Besides saturated fatty acids, also unsaturated fatty acids, hydroxyl-fatty acids, oxo-fatty acids, alkyl sulfates and linear alkylbenzenesulfonates were found.

Besides its functional groups, the surfactant model compound should be representative in its surface-active behaviour. As described in Sect. 2.1, surfactants can be characterized by their separation factor in water $S_{1i}$ and their pure component surface tension $\sigma_i$. By fitting the binary Eberhart model (Eq. 3) to experimental surface tension data reported in El Haber et al. (2024), $S_{1i}$ and $\sigma_i$ were determined for 76 organic substances, which are shown in Fig. 3 (data is provided in tabular form in supplement Sect. S4 and as a csv file, see code and data availability). In addition to these 76 organic compounds, $S_{1i}$ and $\sigma_i$ of atmospheric samples taken at five different locations were considered. Ekström et al. (2010) and Gérard et al. (2016) measured surface tension isotherms of amphiphilic extracts from filter samples of atmospheric aerosol particles collected at coastal, marine, temperate forest, and tropical forest sites. This experimental surface tension data was fitted with the binary Eberhart model (see supplement Sect. S5). The resulting $S_{1i}$ and $\sigma_i$ values are shown as blue squares in Fig. 3.

To show the sensitivity of the results to the choice of the surfactant model compound, we choose six different model surfactants that cover a broad range of $S_{1i}$ values and two regimes of $\sigma_i$. First we choose SDS as a model surfactant having similar $S_{1i}$ and $\sigma_i$ values to that of the atmospheric samples from Ekström et al. (2010). Since surfactants in SSA have been largely identified as fatty acids (Cochran et al., 2016), we further choose three fatty acids among the model compounds, i.e., propionic acid ($C_3H_6O_2$), valeric acid ($C_5H_{10}O_2$), and oleic acid ($C_{18}H_{34}O_2$). From these, propionic acid has the lowest separation factor ($S_{1i} = 55.3$) and can be considered a weakly surface-active substance. Valeric acid has a moderate separation factor ($S_{1i} = 974.6$) that lies between that of propionic acid and SDS. Oleic acid marks the higher end of the separation factor range with $S_{1i} = 9.9 \times 10^6$ and serves to represent the strong surfactants with numbers 59–65 in Fig. 3 as well as the atmospheric samples from Gérard et al. (2016). With these three fatty acids and SDS, four model compounds were found that cover a broad range of $S_{1i}$ while their $\sigma_i$ value is in a small range of $\pm 4\,\mathrm{mN\,m^{-1}}$ around $\sigma_i = 30\,\mathrm{mN\,m^{-1}}$. To show the influence of $\sigma_i$, two additional compounds with a higher $\sigma_i$ of $\sigma_i = 50 \pm 3\,\mathrm{mN\,m^{-1}}$ were chosen, namely glutaric acid and pinonic acid. Glutaric acid can be considered a weakly surface-active substance having a separation factor similar to that of propionic acid



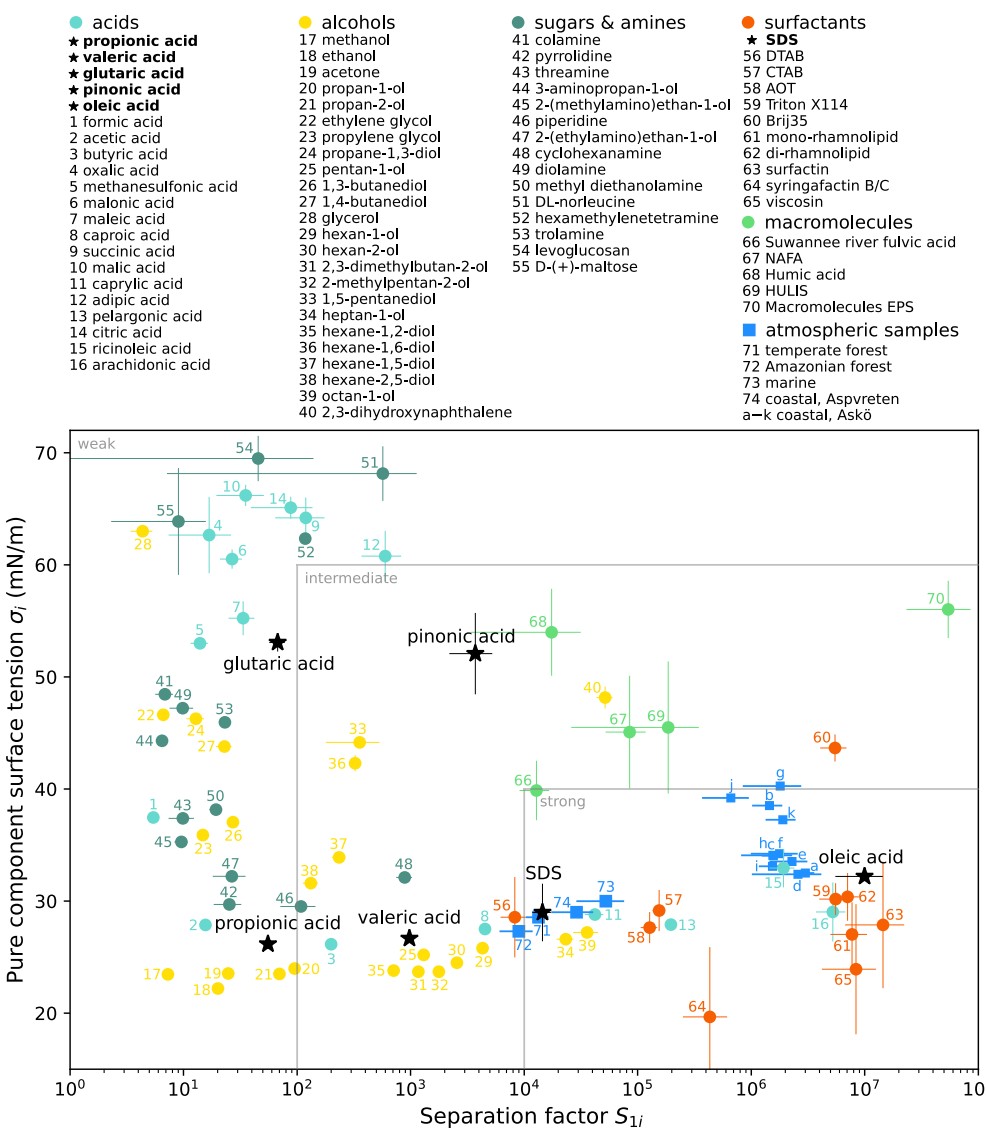

**Figure 3.** Separation factor in water $S_{1i}$ and pure component surface tension $\sigma_i$ of organic substances (stars and 1–70: based on data from El Haber et al. (2024)) and of atmospheric samples taken at 5 different locations (71–74: Ekström et al. (2010), a–k: Gérard et al. (2016), see also supplement Sect. S5). Substances with black stars as markers are used as model compounds in this study. $S_{1i}$ was determined by fitting the binary Eberhart model (Eq. 3) to experimental surface tension data. If $\sigma_i$ was not reported in El Haber et al. (2024), it was considered an additional fitting parameter. For the atmospheric samples, $\sigma_i$ was taken as the lowest measured value. Uncertainty bars show the 95 % confidence intervals of the fit parameters. Substance names and categories are the same as in El Haber et al. (2024), i.e., the category "alcohols" also contains ketones and aldehydes. Grey lines and labels of approximate regions of weak, intermediate, and strong surfactants show the suggested categorization following the results of this study.



($S_{1i} = 67.3$). The separation factor of pinonic acid ($S_{1i} = 3.7 \times 10^3$) lies between that of valeric acid and SDS. Both substances are atmospheric oxidation products and commonly found in secondary organic aerosol. The six model compounds are denoted by black stars in Fig. 3.

    To model the atmospheric surfactants contained in SSA in the most representative manner, sodium dodecyl sulfate (SDS) is chosen as the reference surfactant for three reasons. First, SDS has a surface activity ($S_{1i} = 1.4 \times 10^4$ and $\sigma_i = 29.0\,\mathrm{mN\,m^{-1}}$)

close to the atmospheric surfactants measured by Ekström et al. (2010). Second, its surface tension in mixtures with salts is well characterized (Kleinheins et al., 2023). Third, in agreement with the findings by Cochran et al. (2016), it has a similar molar mass ($M = 288\,\mathrm{g\,mol^{-1}}$) as palmitic acid ($M = 256\,\mathrm{g\,mol^{-1}}$), which is the most abundant species in SSA.

### 3.3   Surface tension parameters of the model compounds

    The multi-component Eberhart model requires $\sigma_i$ of all substances and separation factors $S_{ij}$ between all substance pairs $i$–$j$ in

the mixture. To take surface tension non-ideality into account, additional parameters $A_{ij}$ and $B_{ij}$ are required. In the following, the substances are numbered as (1) water, (2) surfactant, (3) glucose, and (4) NaCl in the subscripts (Fig. 1).

    The pure component surface tension $\sigma_i$ and the binary separation factor in water $S_{1i}$ of all substances used in this study are summarized in Table 2 (see supplement Sect. S6 for underlying experimental data and model fits). For water, $\sigma_1 =$
$72.0\,\mathrm{mN\,m^{-1}}$ was used throughout the study. For the six model substances (propionic acid, glutaric acid, valeric acid, pinonic

acid, SDS, and oleic acid) used in the surfactant category (2), $\sigma_2$ and $S_{12}$ are taken from the fits underlying Fig. 3. The pure liquid surface tension of glucose at room temperature is not known, since glucose crystallizes at this temperature. The binary aqueous solution data from El Haber et al. (2024) is only available in a narrow concentration range. An extrapolation to supersaturated concentrations with the Eberhart model yields a very high value ($\sigma_3 > 10000\,\mathrm{mN\,m^{-1}}$) with a large uncertainty (90 % confidence interval: $\pm 10^7$). In contrast, for sucrose, an extrapolation of aqueous solution data with various models re-

sulted in $\sigma_{\mathrm{sucrose}}$ between $80\,\mathrm{mN\,m^{-1}}$ and $120\,\mathrm{mN\,m^{-1}}$ (Kleinheins et al., 2023), which seems more plausible. Because of the structural similarity between glucose and sucrose, we set the surface tension of glucose to $\sigma_3 = 100\,\mathrm{mN\,m^{-1}}$, i.e., in the middle of the extrapolated range for sucrose. Fitting the Eberhart model with this value to the aqueous solution data for glucose leads to $S_{13} = 1.81$, which means that glucose has a slight tendency to partition to the surface. For NaCl, $\sigma_4 = 169.7\,\mathrm{mN\,m^{-1}}$ and $S_{14} = 0.848$ was used following Kleinheins et al. (2023). The separation factor $S_{14} < 1$ describes a slight depletion of

NaCl at the surface in binary aqueous solution.

    Solute–solute separation factors ($S_{23}$, $S_{24}$, and $S_{34}$) become only important at high solute concentration and are often poorly constrained by available ternary solution data (Kleinheins et al., 2024). In atmospheric aerosol particles, high solute concentrations and even supersaturated solutions can be reached at low relative humidity. However, at activation, the particles are dilute. Therefore, the choice of the solute–solute separation factors has only a minor influence on $SS_{\mathrm{crit}}$. Here, we set

$S_{23}$ and $S_{24}$ for all surfactants to a small value ($10^{-15}$), assuming that the surfactant dominates the surface tension over the co-solute glucose or NaCl at low water content. The glucose–NaCl separation factor is set to $S_{34} = 1$.

    In a preliminary calculation, we tested the influence of salting-out on the Köhler curve. For SDS–NaCl, based on ternary solution data, $A_{24}^{\mathrm{SO}} = 22.63$ and $B_{24}^{\mathrm{SO}} = 2.8 \times 10^3$ were found, which are rather high values compared to other ternary solutions





**Table 2.** Physical properties and surface tension parameters used in this study.

| $i$ | Substance | $\rho_i$ | $M_i$ | $v_i N_A$ | $\sigma_i$ | $S_{1i}$ |
|---|---|---|---|---|---|---|
|  |  | $\mathrm{kg\,m^{-3}}$ | $\mathrm{kg\,mol^{-1}}$ | $\mathrm{cm^3\,mol^{-1}}$ | $\mathrm{mN\,m^{-1}}$ |  |
| 1 | water | 997 | 0.018 | 18.05 | 72.0 | 1 |
| 2 | propionic acid | 988 | 0.074 | 74.90 | 26.2 | 55.3 |
| 2 | glutaric acid | 1219 | 0.132 | 108.29 | 53.1 | 67.3 |
| 2 | valeric acid | 934 | 0.102 | 109.21 | 26.7 | 974.6 |
| 2 | pinonic acid | 965 | 0.184 | 190.67 | 52.1 | $3.7 \times 10^3$ |
| 2 | SDS | 1030 | 0.288 | 279.61 | 29.0 | $1.4 \times 10^4$ |
| 2 | oleic acid | 888 | 0.282 | 317.57 | 32.2 | $9.9 \times 10^6$ |
| 3 | glucose | 1301 | 0.180 | 138.36 | 100.0 | 1.81 |
| 4 | NaCl | 2090 | 0.058 | 27.75 | 169.7 | 0.848 |

that have been examined by Kleinheins et al. (2024). Yet, the difference in $SS_{\mathrm{crit}}$ compared to a calculation assuming ideality

($A_{24}^{\mathrm{SO}} = 0$ and $B_{24}^{\mathrm{SO}} = 0$) was found to be very small ($\Delta SS_{\mathrm{crit}} = 0.015\,\%$). Only when setting $B_{24}^{\mathrm{SO}}$ to an artificially high value of $10^5$, a considerable influence ($\Delta SS_{\mathrm{crit}} > 0.04\,\%$) could be found. This case is shown in supplement Sect. S7. Since such a high salting-out factor does not appear realistic for any of the surfactants considered in this study, we conclude that the influence of salting-out on $SS_{\mathrm{crit}}$ is negligible and proceed with the ideal Eberhart model (Eq. 4, $A_{ij} = 0$, $B_{ij} = 0$) in the following calculations.

In Table 2, additionally the parameters required by the Monolayer model, namely the density $\rho_i$, the molar mass $M_i$ and the molar volume $v_i N_A = M_i/\rho_i$ for the model compounds are given. Pure component densities for substances that are liquid at room temperature were taken from Yaws (1999) for propionic acid, valeric acid, and oleic acid and Lemmon et al. (2023) for water. For glutaric acid, pinonic acid, and glucose, the density of the liquid substance at $25\,°C$ was estimated with the E-AIM model (Clegg and Wexler; Girolami, 1994). For NaCl, the density in liquid state was estimated by an extrapolation

from aqueous solution based on the data from Clegg and Wexler (2011). For SDS, the value specified by the manufacturer Merck KGaA has been used (Merck, 2023).

## 4 Results

### 4.1 Köhler curve of a quaternary SSA particle

To determine the critical supersaturation of SSA particles, Köhler curves are constructed by solving bulk–surface partitioning

and evaluating the Köhler equation (Eq. 1) for a range of wet diameters. In Fig. 4, Köhler curves based on three different model setups are shown for a $D_{\mathrm{dry}} = 50\,\mathrm{nm}$ SSA particle. The main assumptions in the three model approaches are summarized in Table 3 and described in the following.





**Figure 4.** Köhler curves calculated with the Eberhart–Monolayer model, assuming no bulk depletion, and with classical Köhler theory for a SSA model particle (SDS–glucose–NaCl) with $D_{\mathrm{dry}} = 50\,\mathrm{nm}$, $w_{\mathrm{org}} = 0.93$ ("med"), and $w_{\mathrm{glu}}/w_{\mathrm{org}} = 0.05$. First row: Köhler curve (solid or dashed lines) with Raoult (dash-dotted lines) and Kelvin effect (dotted lines) and critical supersaturation (circles). Second row: droplet surface tension. Third row: bulk composition (first column) and total composition (second and third column). The y-axis range was limited to 0–0.02 for a better visibility of the solute share. Fourth row: surface composition in the Eberhart–Monolayer model. Since no partitioning is calculated in the second and third column, the surface composition is not determined and hence not shown here.





**Table 3.** Summary of the assumptions in the three model setups to calculate the bulk mole fractions $x_i^{\mathrm{bulk}}$, the surface tension $\sigma$ and the water activity $a_w$. For details about the calculation of $x_i^{\mathrm{bulk}}$, $\sigma$, and $a_w$ see Sect. 2.

| Model name | $x_i^{\mathrm{bulk}}$ | $\sigma$ | $a_w$ |
|---|---|---|---|
| Eberhart–Monolayer model | $\mathrm{Monolayer}(x_i^{\mathrm{tot}}, D_{\mathrm{wet}})$ | $\mathrm{Eberhart}(x_i^{\mathrm{bulk}})$ | $\mathrm{AIOMFAC}(x_i^{\mathrm{bulk}})$ |
| No bulk depletion | $x_i^{\mathrm{tot}}$ | $\mathrm{Eberhart}(x_i^{\mathrm{tot}})$ | $\mathrm{AIOMFAC}(x_i^{\mathrm{tot}})$ |
| Classical Köhler theory | $x_i^{\mathrm{tot}}$ | $\sigma_1$ | $\hat{x}_w^{\mathrm{tot}}$ |

In the first column, the multi-component Eberhart model (Sect. 2.1) is combined with the Monolayer model (Sect. 2.2) to take bulk–surface partitioning and the surface tension of the partitioned droplet at the specific dilution into account. This model setup is labelled **"Eberhart–Monolayer model"** and represents our best estimate of the real $SS$ of the SSA particles. The composition of the bulk and surface phases of the equilibrated, partitioned droplet are shown in the third and fourth row of Fig. 4, respectively. At small wet diameters the droplet has little total water content leading to a highly concentrated bulk phase and a surface phase that is mostly composed of surface-active SDS. With further dilution at larger wet diameters, the solute concentration in the bulk phase decreases. This leads to a re-partitioning of SDS and a decrease of its surface coverage. As NaCl and glucose are not surface-active substances, they remain mainly in the bulk and cannot be seen in the surface composition. Only when the glucose fraction is high and no strong surfactant is present, some glucose partitions to the surface, as shown in Fig. S9 in supplement Sect. S8. The surface tension of the partitioned droplet (second row) is directly related to the bulk composition via the Eberhart model (Eq. 4 and 9) and to the surface composition via Eq. 10. At low wet diameters where the surface is fully composed of SDS, the surface tension of the droplet is equal to that of pure SDS ($\sigma = \sigma_2$). As the droplet further dilutes with increasing wet diameter, the surface tension increases. Since SDS has a large separation factor, even for an increase in diameter by a factor of 10 ($D_{\mathrm{wet}} = 0.5\,\mathrm{\mu m}$) the surface tension of the droplet is still lower than that of pure water. If a compound with a smaller separation factor is used instead, e.g., propionic acid, the surface phase is less enriched in that compound for a given wet diameter resulting in a higher surface tension such that at activation $\sigma \approx \sigma_1$ (see Fig. S10 in supplement Sect. S8). The surface tension directly affects the Kelvin curve (exponential function in Eq. 1) which is shown as a dotted line in the first row. The Raoult effect ($a_w$ in Eq. 1, shown as a dash-dotted line in the first row), is calculated from $x_i^{\mathrm{bulk}}$ using AIOMFAC (Sect. 2.3). With the Kelvin and the Raoult effect, the Köhler curve can be calculated (Eq. 1, black solid line in the first row) and the critical supersaturation determined ($SS_{\mathrm{crit}} = 0.53\,\%$, black circle).

To illustrate the effect of bulk depletion, in the second column the Köhler curve is shown for the same model setup as in the first column, but neglecting bulk-depletion (labelled **"No bulk depletion"**). This means that bulk–surface partitioning is not calculated, but instead the total composition $x_i^{\mathrm{tot}}$ is used to calculate the surface tension with the Eberhart model and to calculate the water activity with AIOMFAC ($x_2^{\mathrm{bulk}} = x_2^{\mathrm{tot}}$). In the third row of Fig. 4, it can be seen that $x_2^{\mathrm{bulk}}$ in the Eberhart–Monolayer model is lower than in the "No bulk depletion" case, which is due to the partitioning of the surfactant to the surface and its consequent depletion in the bulk phase. Due to the higher $x_2^{\mathrm{bulk}}$ in the second column, the Eberhart model predicts a





lower surface tension, as can be seen in the second row. Additionally, the higher $x_2^{\mathrm{bulk}}$ leads to a slightly lower $a_w$. Both effects

contribute to a lower critical supersaturation ($SS_{\mathrm{crit}} = 0.32\,\%$) than with bulk depletion.

In the third column of Fig. 4, for comparison, Köhler curves are constructed assuming **classical Köhler theory**. In this case, no bulk–surface partitioning is calculated, such that $x_2^{\mathrm{bulk}}$ is identical to that in the second column. In contrast to the "No bulk depletion case", the surface tension of the droplet is assumed to be that of pure water at all wet diameters ($\sigma = \sigma_1$) and solution ideality is assumed ($a_w = \hat{x}_w^{\mathrm{tot}}$). The consistently high surface tension increases the Kelvin effect, which leads to a higher

$SS_{\mathrm{crit}}$ compared to the first two model setups ($SS_{\mathrm{crit}} = 0.98\,\%$). The difference in the Raoult effect, once using AIOMFAC (second column) and once assuming ideality (third column) is small. The reason is that in the case shown in Fig. 4, the droplet is rather dilute at activation such that the assumption of an ideal solution is valid. However, when the organic fraction is high, the droplet can still be concentrated at activation and present in a liquid–liquid phase separated state. In this case, the assumption of ideality leads to a too low Raoult effect compared to the case considering solution non-ideality and LLPS. Two of such cases

are shown in supplement Sect. S3 (Fig. S3, SDS, "high", 50 nm and Fig. S4, pinonic acid, "high", 100 nm). Overall, it can be observed that surface tension lowering leads to a strong reduction in $SS_{\mathrm{crit}}$ in a 50 nm SSA particle.

### 4.2 Influence of the organic content

To analyze the sensitivity of $SS_{\mathrm{crit}}$ on the organic-to-inorganic ratio, in Fig. 5 $SS_{\mathrm{crit}}$ is shown for $w_{\mathrm{org}}$ ranging from $0 - 1$ for 50 nm and 100 nm particles composed of SDS and NaCl. In classical Köhler theory (open circles), since the surface tension

is not a function of composition ($\sigma = \sigma_1$), the Kelvin effect is independent of the composition, too. The strong increase in $SS_{\mathrm{crit}}$ with increasing $w_{\mathrm{org}}$ is therefore due to a change in the Raoult effect. Since SDS has a higher molar volume than NaCl (Table 2) and does not dissociate (van't Hoff factor $= 0$), an SDS particle of a given dry volume results in a smaller number of solute molecules than an NaCl particle of the same dry volume. This leads to a higher $a_w$ and explains the strong increase in $SS_{\mathrm{crit}}$ with increasing $w_{\mathrm{org}}$ predicted using classical Köhler theory. Conversely, using the Eberhart model and neglecting bulk

depletion (grey filled circles) strongly overestimates the surface tension lowering such that the critical supersaturation falls even below the one of pure NaCl for organic contents up to $w_{\mathrm{org}} \approx 0.8$ for $D_{\mathrm{dry}} = 50$ nm.

The Eberhart–Monolayer model (black filled circles in Fig 5) is in between these extreme cases. The increase in the Raoult effect with increasing $w_{\mathrm{org}}$ dominates over the decrease in the Kelvin effect such that pure NaCl activates at a lower supersaturation than any mixed SDS–NaCl particle. Yet, for $w_{\mathrm{org}} \lesssim 0.3$, a mixed SDS–NaCl particle still activates similarly well

as a pure NaCl particle. Between $w_{\mathrm{org}} = 0.3$ and $0.97$, the higher the SDS content, the larger the difference between $SS_{\mathrm{crit}}$ of pure NaCl and the mixed SDS-NaCl particles becomes. For $w_{\mathrm{org}} \gtrsim 0.97$, a strong increase in $SS_{\mathrm{crit}}$ is observed. In this case, due to the hydrophobic nature of SDS and the small amount of NaCl, the hygroscopic growth of the particle is effectively insignificant, leading to the very high $SS_{\mathrm{crit}}$ values.

For 100 nm particles, the results are similar but $SS_{\mathrm{crit}}$ is generally shifted to lower values due to the lower Kelvin effect of

larger particles. Because of the smaller relative importance of the Kelvin effect compared to the Raoult effect, surface tension lowering is also less important leading to a smaller difference between $SS_{\mathrm{crit}}$ calculated with classical Köhler theory and with the Eberhart–Monolayer model. Comparing $SS_{\mathrm{crit}}$ from classical Köhler theory with the one from the Eberhart–Monolayer





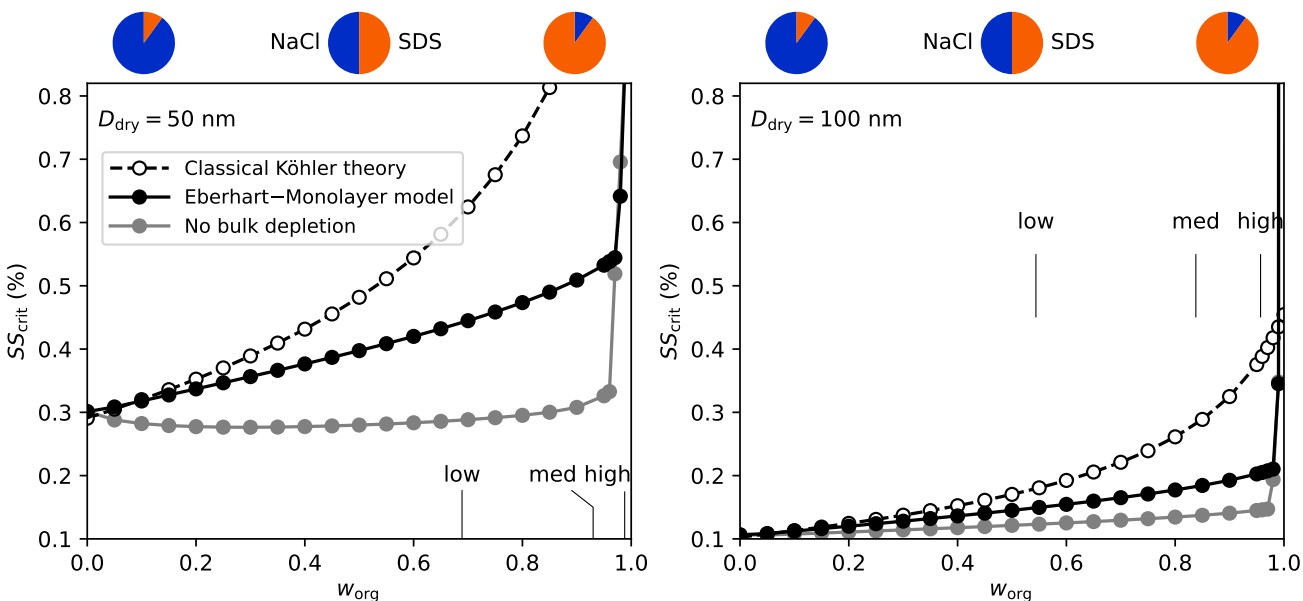

**Figure 5.** Influence of $w_{\mathrm{org}}$ on $S_{\mathrm{crit}}$ for SDS–NaCl particles using three different model approaches. Left panel: $D_{\mathrm{dry}} = 50\,\mathrm{nm}$, right panel: $D_{\mathrm{dry}} = 100\,\mathrm{nm}$. Pie charts above the plots show the composition of the dry particle (in mass fraction) at $w_{\mathrm{org}} = 0.1$, $0.5$, and $0.9$. The annotations "low", "med", and "high" refer to the cases of the same name shown in Fig. 2.

model, it can be seen that there is yet a significant influence of surface tension lowering for $w_{\mathrm{org}} > 0.65$ with differences in $SS_{\mathrm{crit}} \gtrapprox 0.05\,\%$.

### 4.3 Influence of the glucose content

The fraction of WSOC in the total organic mass was found to be small ($\approx 5\,\%$) for small particles according to Facchini et al. (2008). However, atmospheric SSA may exhibit natural variability in its composition. Therefore, we investigate the sensitivity of $SS_{\mathrm{crit}}$ to $w_{\mathrm{glu}}/w_{\mathrm{org}}$ using the three models described in the previous section. In Fig. 6 the result is shown for a $D_{\mathrm{dry}} = 50\,\mathrm{nm}$ SSA particle (left panel) and a $D_{\mathrm{dry}} = 100\,\mathrm{nm}$ SSA particle (right panel) both with medium organic content.

In classical Köhler theory, the decrease in $SS_{\mathrm{crit}}$ with increasing $w_{\mathrm{glu}}/w_{\mathrm{org}}$ is due to a decrease in the Raoult effect due to the lower molar volume of glucose than SDS (see Table 2). When taking surface tension lowering into account, the presence of SDS leads to a lowered surface tension which lowers the Kelvin effect. From the "No bulk depletion" calculations, it can be seen that the lowered Kelvin effect outweighs the increase in the Raoult effect, such that here $SS_{\mathrm{crit}}$ increases with increasing $w_{\mathrm{glu}}/w_{\mathrm{org}}$.

Considering bulk–surface partitioning in the Eberhart–Monolayer model lowers the bulk concentration of SDS and increases the surface tension compared to the "No bulk depletion" case (compare Fig. 4). As a consequence, the Raoult part of the Köhler equation increases due to a higher $a_w$ and the Kelvin part increases due to the higher surface tension. Therefore, $SS_{\mathrm{crit}}$





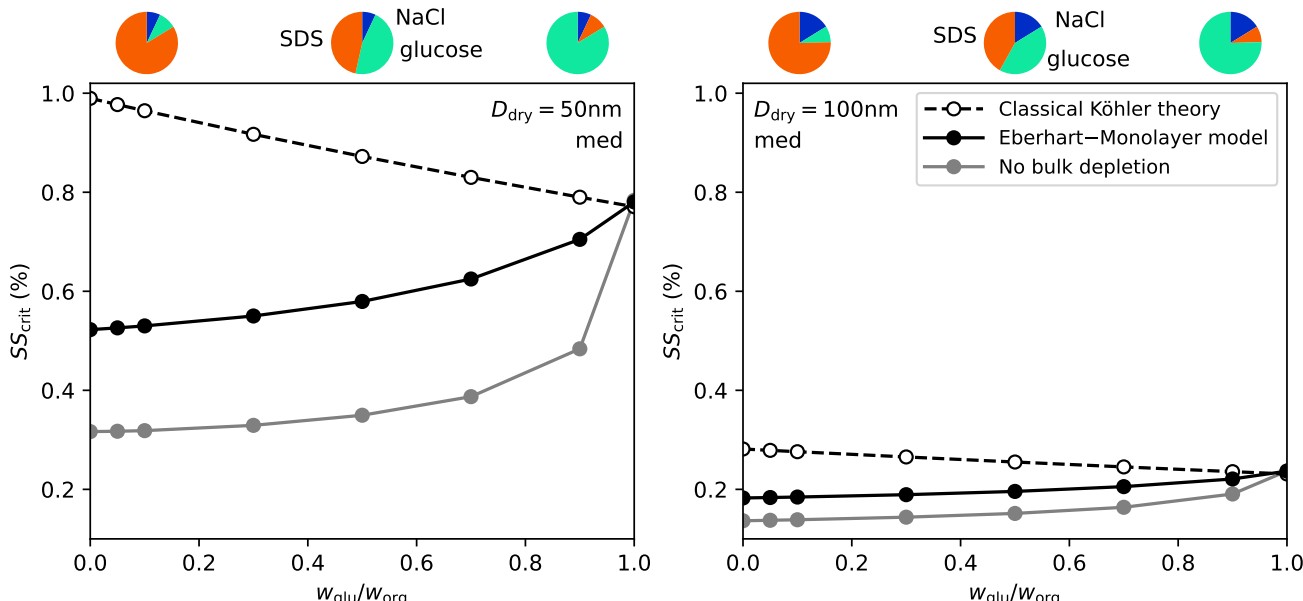

**Figure 6.** Influence of $w_{\mathrm{glu}}/w_{\mathrm{org}}$ on $S_{\mathrm{crit}}$ for SSA model particles (SDS–glucose–NaCl) with three different model approaches. Left panel: $D_{\mathrm{dry}} = 50$ nm, $w_{\mathrm{org}} = 0.93$ ("med"), right panel: $D_{\mathrm{dry}} = 100$ nm, $w_{\mathrm{org}} = 0.838$ ("med"). Pie charts above the plots show the composition of the dry particle (in mass fraction) at $w_{\mathrm{glu}}/w_{\mathrm{org}} = 0.1$, 0.5, and 0.9.

calculated with the Eberhart–Monolayer model is higher than suggested by the "No bulk depletion" case at all $w_{\mathrm{glu}}/w_{\mathrm{org}} < 1$. Yet, similar to the "No bulk depletion" case, the lowered Kelvin effect outweighs the increase in the Raoult effect, leading to an $SS_{\mathrm{crit}}$ that increases with increasing $w_{\mathrm{glu}}/w_{\mathrm{org}}$.

These trends are similar for the calculations with $D_{\mathrm{dry}} = 100$ nm, but all curves are shifted to lower supersaturations because of the strong dependence of the Kelvin effect on the diameter. Overall, in both panels the Eberhart–Monolayer model predicts the lowest $SS_{\mathrm{crit}}$ to be reached at $w_{\mathrm{glu}}/w_{\mathrm{org}} = 0$. At $w_{\mathrm{glu}}/w_{\mathrm{org}} = 1$ all models yield very similar $SS_{\mathrm{crit}}$ values, because in a glucose–NaCl–water droplet, bulk–surface partitioning is very weak at activation such that $\sigma \approx \sigma_1$ in agreement with classical Köhler theory (see Fig. S11 in supplement Sect. S8).

## 4.4 Influence of surfactant properties

To illustrate the influence of the surfactant properties, i.e., $S_{12}$ and $\sigma_2$ on the results, the calculations shown in the left panel of Fig. 6 were repeated with six different surfactant model compounds. The results are shown in Fig. 7 with a different surfactant model compound in each panel, sorted by $S_{12}$ and $\sigma_i$ as in Fig. 3.

We start with analyzing the lower row, which shows results of the substances with $\sigma_i \approx 30 \,\mathrm{mN\,m^{-1}}$. The panel of SDS shows the same results as the left panel in Fig. 6 and has been discussed above. If SDS is replaced by a surfactant with a higher separation factor, i.e., oleic acid, the observed trends are similar and even more pronounced. Propionic acid and valeric acid are





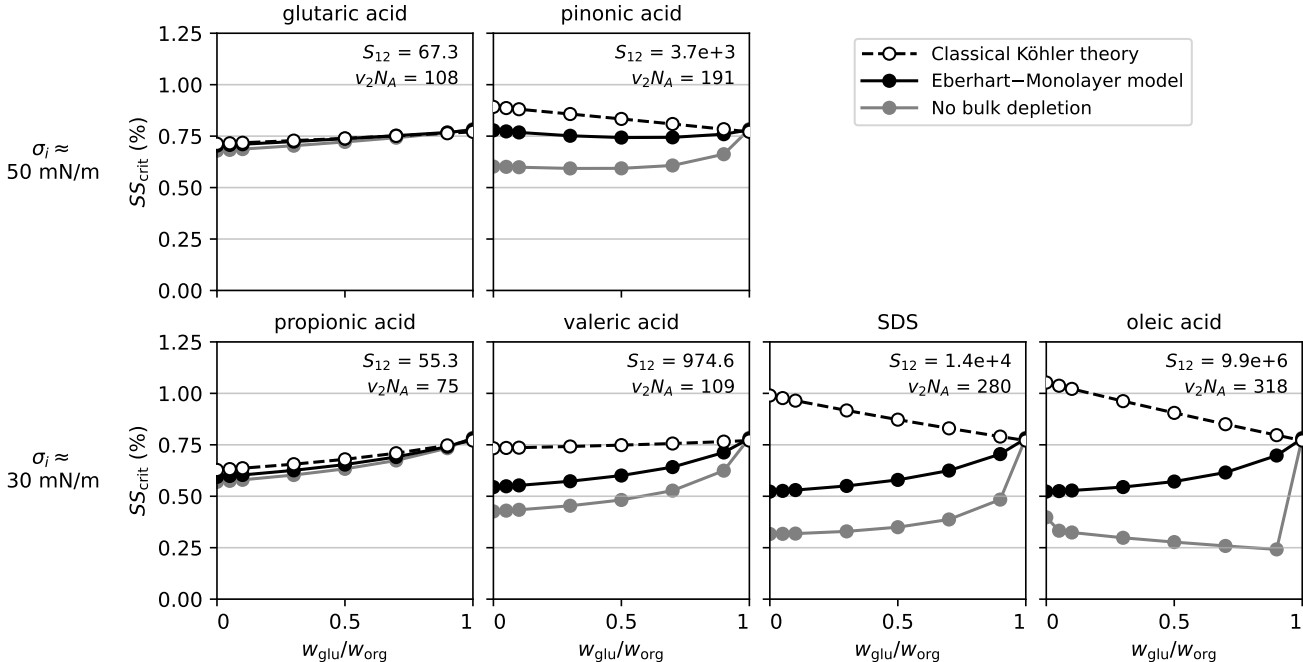

**Figure 7.** Influence of the surfactant type on SSA model particles (surfactant–glucose–NaCl): Critical supersaturation as a function of the glucose content using six different surfactant model compounds and three different model approaches. In all panels, $D_{\text{dry}} = 50\,\text{nm}$ and $w_{\text{org}} = 0.93$ ("med"). The binary separation factor in water $S_{12}$ and the molar volume $v_2 N_A$ in $\text{cm}^3\,\text{mol}^{-1}$ for each surfactant are given in the respective panel. In the calculations with glutaric acid all curves overlap.

substances with a similar $\sigma_2$ as SDS and oleic acid, but with a lower $S_{12}$. For these substances, the differences in $SS_{\text{crit}}$ between the different model approaches decrease with decreasing separation factor and decreasing molar volume. The primary reasons

410 for this are the decrease in $SS_{\text{crit}}$ calculated with classical Köhler theory and the increase of $SS_{\text{crit}}$ for the model approach "no bulk depletion". In contrast, $SS_{\text{crit}}$ from the Eberhart–Monolayer calculations shows little change. In fact, $SS_{\text{crit}}$ calculated with the Eberhart–Monolayer model (black circles) are very similar between propionic acid, valeric acid, SDS, and oleic acid. Because propionic acid and valeric acid have a weaker tendency to partition to the surface, the surface tension at activation is closer to that of pure water which increases the Kelvin effect. Conversely, the Raoult effect for propionic and valeric acids is

415 stronger than for SDS and oleic acid, and can compensate the increased Kelvin effect (detailed plots are shown in Fig. S12 in supplement Sect. S9). Because the molar volumes of valeric acid and glucose are almost the same, $SS_{\text{crit}}$ predicted by classical Köhler theory is almost independent of $w_{\text{glu}}/w_{\text{org}}$ for this model compound. The results obtained with classical Köhler theory depend strongly on the surfactant type because the stronger Raoult effect is not compensated by an increased Kelvin effect since $\sigma = \sigma_1$ for all compositions. For propionic acid, classical Köhler theory predictions of $SS_{\text{crit}}$ are almost the same as

420 the ones of the Eberhart–Monolayer model, so we conclude that it should not be considered a surface-active compound with





respect to CCN activation because of its low separation factor. In contrast, the $SS_{\mathrm{crit}}$ of valeric acid is decreased by surface tension lowering, but less than for SDS, and thus it can be considered a moderately surface-active compound.

Now, we consider glutaric acid and pinonic acid, two substances with $\sigma_2 \approx 50\,\mathrm{mN\,m^{-1}}$. For glutaric acid, classical Köhler theory is sufficient to describe $SS_{\mathrm{crit}}$ at all compositions. Due to its low $S_{12}$ and moderate $\sigma_2$, it should not be considered a surface-active compound for CCN activation. Pinonic acid, on the other hand, has a $S_{12}$ value that is much higher than the one of glutaric acid, lying between the one of valeric acid and SDS. As a result, for pinonic acid, a slightly lower $SS_{\mathrm{crit}}$ is predicted with the Eberhart–Monolayer model compared to with classical Köhler theory and therefore, it can be considered a moderately surface-active compound, similar to valeric acid. The reduction in $SS_{\mathrm{crit}}$, however, is less pronounced than for valeric acid, which can be attributed to its higher $\sigma_2$ value.

The combined influence of surfactant properties and the organic fraction (low, med, or high) is shown in Fig. S13 in supplement Sect S9. When the organic fraction is lowered, the particle contains more NaCl. Due to its low molar mass and its dissociation into two ions, NaCl strongly enhances the Raoult effect and as a result, all curves are shifted to lower $SS_{\mathrm{crit}}$ values. Vice versa, when the organic fraction is high, $SS_{\mathrm{crit}}$ is generally shifted to higher values. LLPS is observed at high organic fraction and low glucose content ($w_{\mathrm{glu}}/w_{\mathrm{org}} \leq 0.05$), for pinonic acid, SDS, and oleic acid. This causes $SS_{\mathrm{crit}}$ to be even higher than at $w_{\mathrm{glu}}/w_{\mathrm{org}} = 0.1$ for the calculations with the Eberhart–Monolayer model and in the "No bulk depletion" case (details on the LLPS see Fig. S3 in supplement Sect. S3). This effect is not present with classical Köhler theory since in that case solution ideality is assumed. For the cases, where the droplet exhibits LLPS at activation, the "No bulk depletion" calculation yields the same result as the Eberhart–Monolayer model because of the low dilution. In all other cases, bulk depletion can not be neglected, but the Monolayer model must be used for an accurate calculation of $SS_{\mathrm{crit}}$ (except when $w_{\mathrm{sf}} = 0$).

Figure S13 in supplement Sect. S9 also shows results for $D_{\mathrm{dry}} = 100\,\mathrm{nm}$. As observed in Fig. 6, $SS_{\mathrm{crit}}$ is shifted to lower values due to the smaller Kelvin effect for larger diameters and the differences between the three approaches become smaller. Moreover, as a consequence of the stronger hygroscopic growth of the larger particles, no LLPS is observed.

From the results in Fig. 7 and S13 (supplement) two main conclusions can be drawn. First, bulk depletion should generally be considered if surface-active compounds are present. Second, solutes in an aerosol particle are best described by three properties—the binary separation factor in water $S_{1i}$, the pure component surface tension $\sigma_i$, and the molar volume $v_i N_A$—to determine their influence on the critical supersaturation. According to the results presented here, substances with $S_{1i} < 100$ should not be considered surface-active for CCN activation and classical Köhler theory can be applied. Substances with $S_{1i}$ between about $100$ and $10^4$ can be considered moderately surface-active, as they influence $SS_{\mathrm{crit}}$ by surface tension lowering to a moderate degree. Substances with $S_{1i} > 10^4$ and low $\sigma_i$ can be considered as strongly surface-active compounds that lower $SS_{\mathrm{crit}}$ substantially compared to classical Köhler theory. Based on these findings, we suggest to categorize organic compounds into weak, intermediate, and strong surfactants with respect to CCN activation, as shown by the grey lines in Fig. 3.

### 4.5 CCN activity of sea spray aerosol

In order to analyze the potential of SSA to serve as CCN, we focus on our best estimate representation of SSA by using SDS as the surfactant and $w_{\mathrm{glu}}/w_{\mathrm{org}} = 0.05$ in the following calculations. In Fig. 8, the critical supersaturation of SSA with

medium organic content is given as a function of the dry diameter calculated with either the Eberhart–Monolayer model or with

classical Köhler theory. On a second y-axis, the size-dependent organic mass fraction of the particle is shown in orange. It can

be clearly seen that when considering bulk–surface partitioning and surface tension lowering by using the Eberhart–Monolayer

model, a SSA particle of a given dry diameter activates at lower $SS$ than predicted by classical Köhler theory. The difference

in $SS_{\mathrm{crit}}$ between the two model approaches is larger, the smaller the dry diameter. There are two reasons for that. On the one

hand, smaller particles have a higher organic content and therefore a higher surfactant content, and, on the other hand, smaller

particles have a stronger Kelvin effect, such that a change in the Kelvin effect has a larger effect on $SS_{\mathrm{crit}}$.

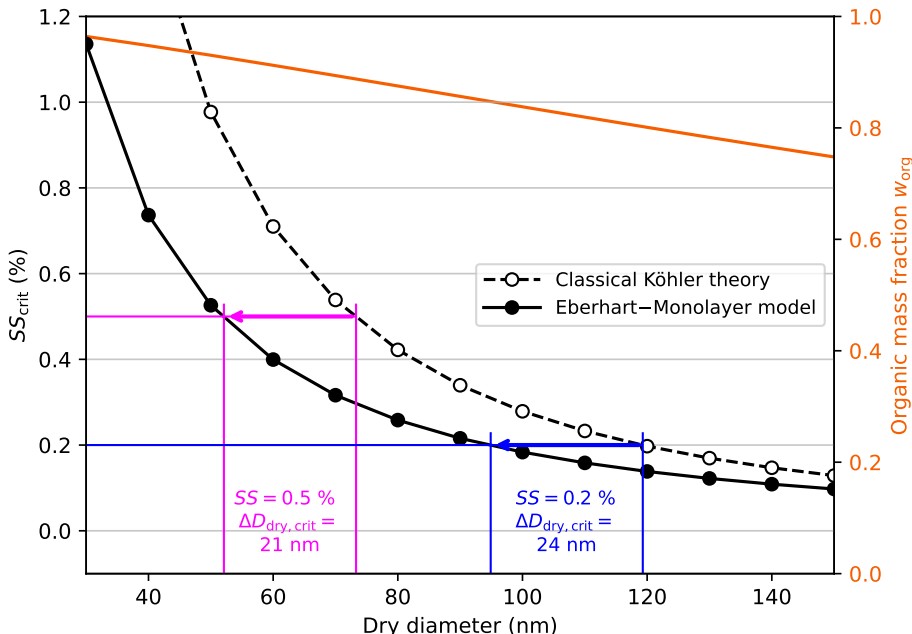

**Figure 8.** The critical supersaturation $SS_{\mathrm{crit}}$ of SSA model particles (surfactant–glucose–NaCl) with medium organic content calculated with

two different model approaches. SSA particles are represented as a quaternary mixture of water, SDS, glucose, and NaCl. The organic mass

fraction (case "med") is given on the right y-axis with an orange solid line. The fraction of glucose in the organic mass is $w_{\mathrm{glu}}/w_{\mathrm{org}} = 0.05$.

The difference in the critical dry diameter $\Delta D_{\mathrm{dry,crit}}$ between the models for a supersaturation $SS$ of $0.5\,\%$ and $0.2\,\%$ are annotated in

magenta and blue respectively.

     The critical dry diameter $D_{\mathrm{dry,crit}}$ denotes the dry diameter of the smallest particles in a polydisperse aerosol of a uniform

composition that can activate at a given supersaturation. Assuming equilibrium in an air parcel at all times, $D_{\mathrm{dry,crit}}$ is given

by $SS = SS_{\mathrm{crit}}$. In Fig. 8 it can be seen that for a supersaturation of $SS = 0.5\,\%$, surface tension lowering yields a $D_{\mathrm{dry,crit}}$

that is $21\,\mathrm{nm}$ lower than that predicted by classical Köhler theory. Analogously, surface tension lowering leads to a change

in $D_{\mathrm{dry,crit}}$ of $24\,\mathrm{nm}$ at a supersaturation of $SS = 0.2\,\%$. For SSA with a low organic content, the change in $D_{\mathrm{dry,crit}}$ is

smaller ($\Delta D_{\mathrm{dry,crit}}(S = 0.5\,\%) = 10\,\mathrm{nm}$ and $\Delta D_{\mathrm{dry,crit}}(S = 0.2\,\%) = 11\,\mathrm{nm}$) and, vice versa, for SSA with a high organic





content the change in $D_{\mathrm{dry,crit}}$ is higher ($\Delta D_{\mathrm{dry,crit}}(S = 0.5\,\%) = 29\,\mathrm{nm}$ and $\Delta D_{\mathrm{dry,crit}}(S = 0.2\,\%) = 41\,\mathrm{nm}$). These cases are shown in supplement Sect. S10.

In marine environments, supersaturations of $SS = 0.5\%$ can be reached (Gong et al., 2023; Svensmark et al., 2024). Based on the results in Fig. 8, this means that due to surface tension lowering, a large part of the Aitken mode particles can be activated ($D_{\mathrm{dry,crit}}(SS = 0.5\,\%) \approx 50\,\mathrm{nm}$) despite having a high organic content.

The CCN activity of SSA particles and the importance of surface tension lowering can further be illustrated by comparing $SS_{\mathrm{crit}}$ of SSA particles to that of pure components. In panel (A) of Fig. 9, the $SS_{\mathrm{crit}}$ of SSA calculated with classical Köhler

theory and with the Eberhart–Monolayer model are shown together with the $SS_{\mathrm{crit}}$ of pure glutaric acid, ammonium sulfate ($(\mathrm{NH_4})_2\mathrm{SO_4}$) and NaCl. In panel (B), the difference between $SS_{\mathrm{crit}}$ of the various cases and the one predicted with the Eberhart–Monolayer model are shown. Compared to Fig. 8, the range of dry diameters is extended here up to $1\,\mathrm{\mu m}$ in both panels. For the pure components, AIOMFAC was used for the calculation of the water activity and a constant surface tension of $\sigma_1$ was used. The prediction of $SS_{\mathrm{crit}}$ of SSA particles with classical Köhler theory (black dashed line) changes from being

higher than the one for glutaric acid (at $D_{\mathrm{dry}} < 100\,\mathrm{nm}$) to lower than ammonium sulfate (at $D_{\mathrm{dry}} > 300\,\mathrm{nm}$), which is related to the change in the organic content of SSA with size. If surface tension lowering is taken into account with the Eberhart–Monolayer model, a CCN activity of SSA similar to the one of pure $(\mathrm{NH_4})_2\mathrm{SO_4}$ is found at $D_{\mathrm{dry}} \lesssim 300\,\mathrm{nm}$. For $D_{\mathrm{dry}} > 300\,\mathrm{nm}$, the CCN activity approaches that of NaCl and classical Köhler theory is valid. For SSA particles with $D_{\mathrm{dry}} \lesssim 60\,\mathrm{nm}$ and high organic content, LLPS causes $SS_{\mathrm{crit}}$ to be closer to that of glutaric acid than that of $(\mathrm{NH_4})_2\mathrm{SO_4}$.

In climate models, parametrizations for CCN activation are usually used, which are based on $\kappa$-Köhler theory (Petters and Kreidenweis, 2007). In this theory, the water uptake and $SS_{\mathrm{crit}}$ of aerosol particles is calculated using classical Köhler theory and a hygroscopicity parameter $\kappa$ which considers the molar volumes and the degree of dissociation of the solutes. For particles composed of various solutes, the overall $\kappa$ value of the particle is calculated from the individual $\kappa$ values of the solutes using the Zdanovskii–Stokes–Robinson (ZSR) rule (Petters and Kreidenweis, 2007). Organic substances are typically given low $\kappa$ values

(e.g., $\kappa < 0.06$ (Zhang et al., 2012)) or assumed to be entirely hydrophobic ($\kappa = 0$). The black dashed lines in Fig. 9 correspond $\kappa \approx 0.06$ in $\kappa$-Köhler theory based on the molar volume of SDS. In addition, the black dotted lines show the calculation where the surfactant is assumed to be entirely hydrophobic (see also the underlying Köhler curves given by the yellow dashed lines in Fig. S3 and S4 in supplement Sect. S3), which corresponds to choosing $\kappa = 0$. In Fig. 9 (B) it can be seen that both cases using classical Köhler theory lead to a strong underestimation of the activation potential of SSA particles and are less suited to

represent their $SS_{\mathrm{crit}}$ than assuming pure NaCl.

## 5    Discussion

### 5.1    Uncertainties in the modelling approach

The results of this study showed that classical Köhler theory strongly underestimates the ability of surfactant-rich particles to form cloud droplets and that instead a more complex approach including concentration dependent surface tension and bulk-

depletion is required. It was furthermore shown that SSA particles in the Aitken mode with diameters down to $\approx 50\,\mathrm{nm}$ can



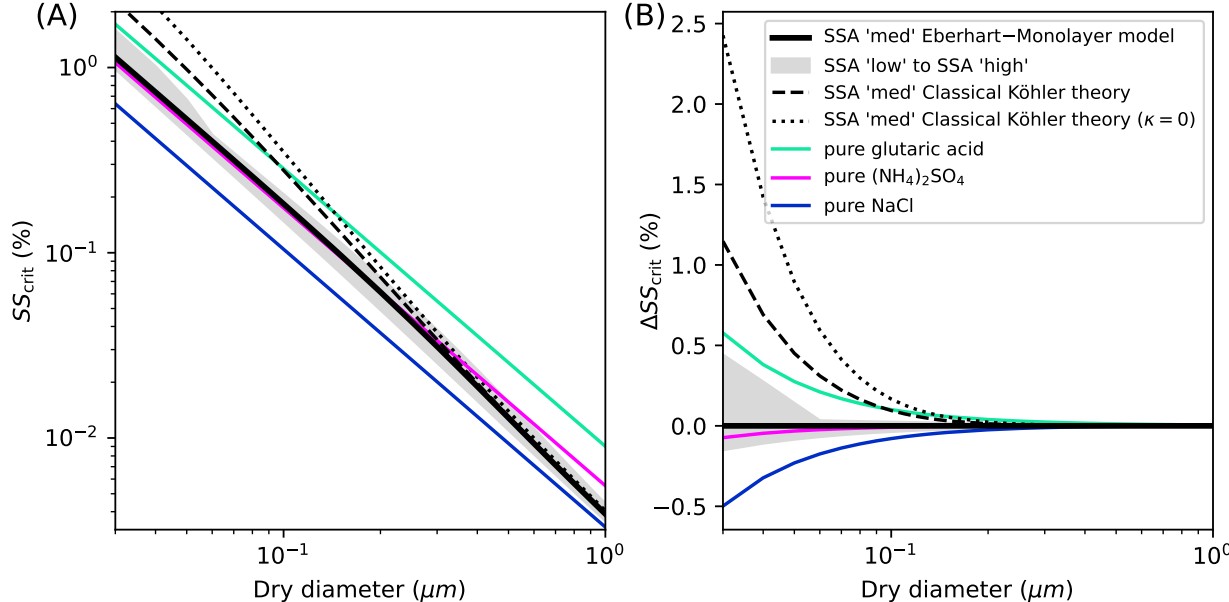

**Figure 9.** Comparison of the CCN activation potential of SSA model particles (surfactant–glucose–NaCl) with pure compounds. (A): critical supersaturation as a function of dry diameter. (B): difference to calculation with the Eberhart–Monolayer model. The curves "SSA 'med' Classical Köhler theory" and "SSA 'med' Eberhart–Monolayer model" are identical to those shown in Fig. 8 up to 150 nm. The gray shaded area shows the range between the $SS_{crit}$ of SSA with low and high organic content, as predicted with the Eberhart–Monolayer model.

serve as CCN particles in typical marine updrafts. These results are based on the assumptions of SSA composition, the choice of model compounds, and the modelling approach, which all influence the resulting $SS_{crit}$ for CCN activation. The SSA particles have been represented with four model compounds (surfactant, glucose, NaCl, and water) and with a size dependent composition based on SSA measurements. Despite the simplicity of this model system, we consider our results robust, even if the composition of atmospheric SSA has not been accurately represented. The reasons for that are discussed in the following.

the composition of atmospheric SSA has not been accurately represented. The reasons for that are discussed in the following.

The biggest uncertainty in the presented model approach might be the organic-to-inorganic ratio of SSA particles; especially the organic-to-inorganic ratio of particles with $D_{dry} < 100\,\mathrm{nm}$ is uncertain due to a lack of experimental data. To analyze the sensitivity to this parameter, we have shown results with low and high organic content in addition to the medium organic content. The resulting variation of $SS_{crit}$ is shown in Fig. 9 with the gray shaded area. It can be seen that even for a lower or

higher inorganic content, $SS_{crit}$ is still close to our best estimate (black line). A sensitivity analysis over the whole range of organic-to-inorganic ratios (Fig. 5) showed that only if the organic content of small particles is below approximately $\approx 40\,\%$, the influence of surface tension lowering on $SS_{crit}$ becomes insignificant and classical Köhler theory suffices. In this case, however, the particles would activate similar to pure NaCl thus being good CCN anyhow. We therefore consider it robust to



conclude that SSA particles in the Aitken mode can serve as CCN particles in marine updrafts, regardless of whether this is
due to surface tension lowering at high organic content or due to a strong hygroscopic growth at high inorganic content.

The WSOC content could depend on seasonal and spatial variability and aging via the uptake of organic molecules from
the gas phase or chemical breakdown of large surface-active organic molecules. Yet, in Fig. 6 and 7 it has been shown that the
glucose content between 0 and 0.5 has only a small influence on $SS_{\mathrm{crit}}$ calculated with the Eberhart–Monolayer model.

The surfactants in atmospheric SSA are composed of a complex mixture of different compounds, which we represented with
pure SDS. The molecular identification of surfactants in SSA is still at an early stage and should clearly be investigated further.
Nevertheless, there is increasing evidence that atmospheric surfactants are predominantly composed of fatty acids. Exchanging
SDS with a fatty acid of small, medium or large chain length had little influence on $SS_{\mathrm{crit}}$ calculated with the Eberhart–
Monolayer model (compare results for propionic acid, valeric acid, SDS, and oleic acid in Fig. 7), possibly because these
substances all have a similar $\sigma_i$ of $\approx 30\,\mathrm{mN\,m^{-1}}$. When a substance with a higher $\sigma_i$ was chosen instead, the surface tension
lowering was less pronounced and $SS_{\mathrm{crit}}$ was found to be higher compared to the cases with fatty acids (see glutaric acid and
pinonic acid in Fig. 7). Therefore, if surfactants in atmospheric SSA were found to have $\sigma_i >> 30\,\mathrm{mN\,m^{-1}}$, the conclusions of
this study would need to be reconsidered. Since all fatty acids shown in Fig. 3 have $\sigma_i \approx 30\,\mathrm{mN\,m^{-1}}$, we consider our results
based on SDS as a representative surfactant species to be robust.

Finally, all model calculations have been performed at $25\,^{\circ}\mathrm{C}$, a temperature that is hardly reached at typical liquid cloud
base heights, which range from $200\,\mathrm{m}$ to $2\,\mathrm{km}$ above ground (Lu et al., 2021). Accordingly, CCN activation rather occurs
at temperatures around or below $0\,^{\circ}\mathrm{C}$. To estimate the uncertainty introduced by assuming $25\,^{\circ}\mathrm{C}$ throughout this study, we
performed calculations similar to that shown in Fig. 8 at two different temperatures ($-25\,^{\circ}\mathrm{C}$ and $+25\,^{\circ}\mathrm{C}$) but using oleic
acid instead of SDS. Note that for SDS no temperature dependent surface tension data could be found, which is why oleic
acid was used for this sensitivity analysis. The direct effect of temperature on the Kelvin effect was considered as well as
the indirect effect via the temperature dependence of the surface tension. Both effects lead to an increased Kelvin effect at
lower temperatures, which results in a change of $SS_{\mathrm{crit}}$ to higher absolute values (see Appendix A). Furthermore, at a lower
temperature, the relative difference between $SS_{\mathrm{crit}}$ calculated with classical Köhler theory and the Eberhart–Monolayer model
is slightly increased, suggesting that surface tension lowering could be even more relevant at lower temperatures.

Our results are based on a model approach that combines three pre-existing models: the Eberhart model for surface tension,
the Monolayer model for bulk–surface partitioning, and AIOMFAC for solution non-ideality. All three models have been
validated by comparison to experimental data: The Eberhart model has been thoroughly validated for binary aqueous solutions
and multi-component solutions (Kleinheins et al., 2023, 2024). The Monolayer model has been validated by direct comparison
to surface tension of small droplets (Bzdek et al., 2020; Bain et al., 2023). AIOMFAC has been validated against water activity
measurements (Zuend et al., 2008, 2011). As an additional validation step, future work should be directed at comparing $SS_{\mathrm{crit}}$
predicted with the combined model to measurements of $SS_{\mathrm{crit}}$ of lab-generated or atmospheric SSA aerosol particles. In the
literature, other partitioning models have been suggested besides the Monolayer model, e.g., the Gibbs model (Sorjamaa et al.,
2004; Prisle et al., 2010). Lin et al. (2018) compared these two models for various systems and found that the Monolayer model
generally predicts slightly lower $SS_{\mathrm{crit}}$ than the Gibbs model. Since only few experimental data from $SS_{\mathrm{crit}}$ measurements





were available, no conclusion was drawn as to which model is more accurate. In this study the Monolayer model has been
chosen, since it has been compared to experimental surface tension data of small droplets (Bzdek et al., 2020; Bain et al., 2023)
with good agreement; however, it remains to be tested in future studies whether the Gibbs model would yield a higher accuracy
by validation to the same data. Moreover, more measurements of the surface tension of small droplets, also using different
experimental techniques, would help to elucidate which partitioning model is the most accurate. To conclude, a weaker or an
even stronger bulk depletion effect cannot be excluded, which would shift the predicted critical supersaturation to either lower
or higher values, respectively.

## 5.2 Comparison to field and laboratory measurements

Our results suggest that despite a high organic content, SSA particles can activate similarly well as ammonium sulphate.
Comparing this to CCN measurements of field and laboratory studies in the literature, our results are lying within the range of
what has been measured. Quinn et al. (2014) collected fresh SSA particles in the North Atlantic Ocean with an in situ particle
generator (Sea Sweep). From CCN counter measurements, a rather low CCN activity with $\kappa$ values between 0.4 and 0.8 has
been measured for SSA particles with $D_{\mathrm{dry}} = 40 - 100\,\mathrm{nm}$, which is thought to be due to a high organic content. However, the
organic content has not been measured in that study and a later study of the same two first authors claims that the particles had
not been dried enough in these measurements, which would lead to an overestimation of the size and a low bias for $\kappa$ (Bates
et al., 2020). Rasmussen et al. (2017) and Christiansen et al. (2020) measured the CCN activity of lab generated SSA particles
using seawater samples with and without algae cultures or sea surface microlayer samples. According to their measurements,
the CCN activity of real seawater SSA resembled that of SSA generated from Sigma sea salt water in all cases very closely.
Since the organic content of the SSA after aerosolization has not been quantified, it is not possible to directly compare their
results to ours. Lastly, Bates et al. (2020) measured the CCN activity of SSA particles generated by Sea Sweep and in a tank
and also quantified the organic content of the sub $180\,\mathrm{nm}$ fraction with FTIR measurements. Despite a high average organic
content of $86\,\%$, the particles showed a strong CCN activity in the range between that of pure ammonium sulfate and that of
pure NaCl (Fig. 5 in Bates et al. (2020)), which is close to the results in our study. For a better comparison of model results with
measurements, it is crucial that future studies are undertaken to quantify the organic content of the aerosolized, dried aerosol
that enters the CCN counter.

## 5.3 Representation of CCN activation in global models

In climate models, parametrizations based on classical Köhler theory are usually used to determine the CCN activation of
aerosol particles. For example, in the global circulation model ECHAM-HAM (Stier et al., 2005; Neubauer et al., 2019), cloud
droplet activation is calculated using $\kappa$-Köhler theory (Petters and Kreidenweis, 2007), which is based on classical Köhler
theory. As such, the surface tension of water is generally assumed. The degree of under- or overestimation of CCN activation
in climate models depends further on their representation of the composition. If the marine aerosol is assumed to be composed
entirely of NaCl, the CCN activity for particles $< 100\,\mathrm{nm}$ would be overestimated, i.e., a too low $SS_{\mathrm{crit}}$ would be assumed (see
blue line in Fig. 9 (B)). If, on the other hand, an organic share is considered as for example in the parametrization by Long et al.





(2011), the magnitude of the error depends on the hygroscopicity of the organic mass that is assumed. If the organic mass is given a certain hygroscopicity, e.g., $\kappa \approx 0.06$ based on the molar volume of SDS, the predicted CCN activity would correspond to that predicted by our calculations with classical Köhler theory assuming solution ideality. As such, $SS_{\text{crit}}$ would be strongly
overestimated for all $D_{\text{dry}} \lesssim 300\,\text{nm}$ as shown by the black dashed line in Fig. 9 (B). If the other extreme is considered, that is, the organic share is assumed to be entirely hydrophobic ($\kappa = 0$), the predicted $SS_{\text{crit}}$ is even higher (see black dotted line in Fig. 9 (B)). Using the Eberhart–Monolayer model as the ground truth, Fig. 9 (B) suggests that for the prediction of CCN activation assuming the marine primary aerosol to be composed entirely of pure NaCl is more accurate than considering an organic share with a low $\kappa$ value.

In Sect. 4.5 it has been shown that Aitken mode particles can serve as CCN in marine updrafts. In ECHAM-HAM, however, sea salt aerosol is only included in the accumulation mode ($D_{\text{dry}} = 100\,\text{nm}$–$1\,\mu\text{m}$, Stier et al., 2005). Given the high CCN activation potential of both surfactant-rich and NaCl-rich SSA particles found in this study, ECHAM-HAM seems to underestimate the CCN concentration in marine environments. This finding is in agreement with a study by Lohmann and Leck (2005), where in a parcel model simulation, a surface active Aitken mode had to be included in order to bring the results into
agreement with measured CCN concentrations from field campaigns in the Arctic.

## 6   Summary and conclusions

In this study we have presented an approach for modelling the critical supersaturation of surfactant-containing aerosol particles. The model is based on combining the multi-component Eberhart model (Kleinheins et al., 2024) for the surface tension of aqueous mixtures with the Monolayer model (Malila and Prisle, 2018) for bulk–surface partitioning (Eberhart–Monolayer
model). To consider non-ideality and liquid–liquid phase separation, the group contribution model AIOMFAC was used. This novel model approach allows for the first time to calculate the CCN activity of surfactant containing aerosol particles with more than one co-solute. For comparison, the CCN activity was calculated additionally using classical Köhler theory, which assumes a homogeneous droplet, solution ideality and a surface tension of pure water.

The new model was applied on freshly-emitted sea spray aerosol particles, which were represented by a quaternary system of
water, a surfactant, glucose, and NaCl. Six different model compounds were chosen to represent the surfactant, covering a broad range of surface-active behaviour. The surface-activity of the surfactant model compounds was characterized by their separation factor in water $S_{1i}$ and their pure component surface tension $\sigma_i$. Based on this sensitivity analysis, it can be concluded that substances with a separation factor $S_{1i} < 100$ should not be considered surfactants for CCN activation. For these substances, classical Köhler theory can be used. In contrast, substances with a separation factor between $10^2$ and $10^4$ can be considered
intermediate surfactants, leading to moderate reductions in $SS_{\text{crit}}$ depending on their $\sigma_i$ value while substances with $S_{1i} > 10^4$ and low $\sigma_i$ (e.g., $\sigma_i = 30\,\text{mN}\,\text{m}^{-1}$) should be considered as substances with a strong surface activity. For the latter, strong reductions in $SS_{\text{crit}}$ compared to classical Köhler theory were observed as a result of a strongly reduced Kelvin effect.

Additionally, the critical supersaturation as a function of the dry particle size was presented using either classical Köhler theory or the Eberhart–Monolayer model. The results showed that the CCN activity of sea spray aerosol particles with a



dry diameter below $300\,\mathrm{nm}$ are clearly underestimated when surface tension lowering is not considered. For example, in an updraft with a maximum supersaturation of $0.5\,\%$, surface tension lowering allows SSA particles down to $D_{\mathrm{dry}} \approx 50\,\mathrm{nm}$ to activate while classical Köhler theory predicts CCN activation only for particles with $D_{\mathrm{dry}} \gtrapprox 70\,\mathrm{nm}$. From a comparison of sea spray aerosol with pure compounds this study suggests that their CCN activity is similar to that of ammonium sulfate for dry diameters below $\approx 300\,\mathrm{nm}$. For larger sizes, their CCN activity approaches that of pure NaCl, due to their increasing NaCl

content with size. The $SS_{\mathrm{crit}}$ of SSA is predicted to be closer to that of pure NaCl than to that predicted with $\kappa$-Köhler theory using $\kappa \leq 0.06$. Thus, accounting for an organic fraction in marine primary aerosol may worsen CCN number predictions in climate models compared to neglecting the organic fraction altogether. The low critical activation diameter for SSA particles furthermore emphasizes the need to consider the Aitken mode of SSA particles for cloud formation in climate models.

From this study it can further be concluded that bulk depletion must be considered when particles contain surfactants and

that surface tension non-ideality (i.e., salting-out) is only important at high salt concentrations and not relevant for CCN activation due to the high dilution of the particles at activation. It was further observed that small particles with a high organic content can undergo LLPS due to solution non-ideality, which increases their critical supersaturation above the one obtained when assuming solution ideality. Lastly, a lower temperature was shown to increase the absolute value of $SS_{\mathrm{crit}}$ and slightly increase the relative difference between $SS_{\mathrm{crit}}$ calculated with classical Köhler theory and the one calculated with the Eberhart–

Monolayer model highlighting the importance of surface tension lowering also at lower temperatures.

Overall, this study mapped out the impact of surface tension lowering on the CCN activity of primary sea spray aerosol. Future work could be directed at analyzing the surface tension and CCN activity of secondary marine aerosol or continental aerosol, e.g., biomass burning aerosol.

*Code and data availability.* The results of this study, i.e., critical supersaturations of all systems and the $S_{1i}$ and $\sigma_i$ values of Fig. 3 are

provided at https://doi.org/10.5281/zenodo.13589001. An example code for the Eberhart–Monolayer model to reproduce Fig. S8 and Fig. 4 is provided in the form of a jupyter notebook at https://doi.org/10.5281/zenodo.13588318.

*Author contributions.* JK developed the model code, conducted the data curation, the simulations, the visualizations, and wrote the original draft. CM & UL acquired the funding and administrated the project and supervised JK together with NS. All authors contributed to conceptualization, methodology and writing (review & editing), and all authors have approved the final version of the paper.

*Competing interests.* The authors declare that they have no conflict of interest.

*Acknowledgements.* This research has been funded by the Swiss National Science Foundation (SNF; grant no. 200021L_197149) as part of the ORACLE project. We acknowledge fruitful discussions with Nønne Prisle, Merete Bilde and Barbara Nozière.



**Appendix A: Influence of temperature**

Since surface tension is known to be a function of temperature, we test here the influence of temperature on our results. The
critical supersaturation is a direct function of the temperature $T$ via the Kelvin effect (exponential function in Eq. 1). Besides
the direct influence, $SS_{\mathrm{crit}}$ is indirectly affected by $T$ via a temperature dependence of activity coefficients, densities, and
surface tension. We consider the direct influence of temperature in the Kelvin effect as well as the temperature dependence of
the surface tension in the following and neglect the temperature dependence of activity coefficients and densities.

We introduce temperature dependent surface tensions by replacing $\sigma_i$ in the multi-component Eberhart model (Eq. 4) with
$\sigma_i(T)$ and assume the fit parameters $S_{ij}$ to be temperature independent. Assuming temperature-independent fit parameters
proved to be successful for the Connors-Wright and the Shereshefsky model (Shereshefsky, 1967) by Shardt and Elliott (2017).
For a binary system 1–2, the Shereshefsky model is given by

$$\sigma = \sigma_1 - \frac{(\sigma_1 - \sigma_2)\, x_2 \exp\left(\Delta F_s/(RT)\right)}{1 + x_2 \left(\exp\left(\Delta F_s/(RT)\right) - 1\right)}, \tag{A1}$$

where $\Delta F_{12}$ is a fit parameter. Tahery et al. (2005) showed that, while the fit parameter $\Delta F_{12}$ depends on temperature,
$\Delta F_{12}/(RT)$ is approximately temperature independent for many binary aqueous systems with organic solutes. Since the Eber-
hart model is mathematically equivalent to the Shereshefsky model with $S_{12} = \exp\left(\Delta F_{12}/(RT)\right)$ (Kleinheins et al., 2023), it
can be assumed that $S_{12}$ is also constant with temperature for these systems. Based on these previous studies on temperature
dependence of surface tension, we consider $S_{ij}$ temperature independent.

The temperature dependence of the pure components was modelled with a linear relationship (Shardt and Elliott, 2017) as

$$\sigma_i(T) = \theta_{0,i} + \theta_{1,i} T, \tag{A2}$$

where $\theta_{0,i}$ and $\theta_{1,i}$ are fit parameters specific for compound $i$. No temperature dependent surface tension data could be found
for SDS. However, we could find temperature dependent experimental data for oleic acid, hence we continue this analysis
using oleic acid as the surfactant. The parameters $\theta_{0,i}$ and $\theta_{1,i}$ were determined by fitting Eq. A2 to experimental data reported
by Wohlfarth and Wohlfarth (1997), which is shown in the left panel in Fig. A1. For water, $\theta_{0,i}$ and $\theta_{1,i}$ were taken from
Shardt and Elliott (2017). For glucose, no pure component surface tension data at any temperature could be found, therefore,
its temperature dependence is neglected here. For NaCl, the temperature function by Janz (1980) was used. The parameters
$\theta_{0,i}$ and $\theta_{1,i}$ of these four model compounds are reported in Table A1.

The influence of temperature on $SS_{\mathrm{crit}}$ was analyzed by running a similar calculation as the one behind Fig. 8, yet with oleic
acid instead of SDS and for two temperatures, i.e., $-25\,^{\circ}\mathrm{C}$ and $25\,^{\circ}\mathrm{C}$. The comparison of $SS_{\mathrm{crit}}$ for these temperatures is given
in Fig. A2 and the parameters for the four model compounds at $-25\,^{\circ}\mathrm{C}$ and $25\,^{\circ}\mathrm{C}$ are reported in Table A1. Both the higher
surface tension at lower temperature and the lower temperature itself contribute to a higher Kelvin effect. As a result, $SS_{\mathrm{crit}}$
is higher at a lower temperature for the same dry diameter. Moreover, the relative difference between $SS_{\mathrm{crit}}$ calculated with
Classical Köhler theory and with the Eberhart–Monolayer model is slightly increased, i.e., $\Delta D_{\mathrm{dry,crit}}$ values for $SS = 0.5\,\%$
and $SS = 0.2\,\%$ are larger at $-25\,^{\circ}\mathrm{C}$, because the temperature dependence of the surface tension of water is stronger than the
one of oleic acid (see right panel of Fig. A1).



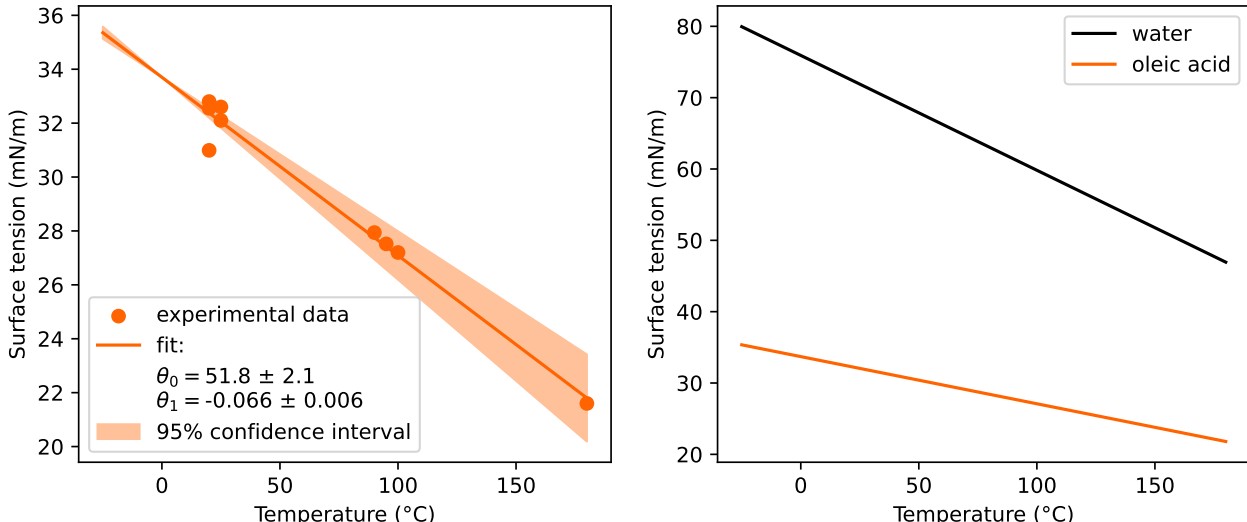

**Figure A1.** Left panel: Surface tension of pure oleic acid as a function of temperature. Markers show experimental data reported by Wohlfarth and Wohlfarth (1997). The solid line is a fit of Eq. A2 to the experimental data resulting in the parameters reported in the legend ($\pm$ 95 % confidence interval). The shaded area gives the 95 % confidence interval of the fit. Right panel: Comparison of the surface tension of pure oleic acid and pure water as a function of temperature using the parameters reported in Table A1.

**Table A1.** Temperature dependence parameters ($\theta_{0,i}$, $\theta_{1,i}$) with references and pure component surface tensions at $-25\,°\mathrm{C}$ and $25\,°\mathrm{C}$ (calculated with $\theta_{0,i}$, $\theta_{1,i}$ and Eq. A2) for the model compounds

| Compound | $\theta_{0,i}$ | $\theta_{1,i}$ | Reference | $\sigma_i(T=-25\,°\mathrm{C})$ | $\sigma_i(T=+25\,°\mathrm{C})$ |
| --- | --- | --- | --- | --- | --- |
| | $(\mathrm{mN\,m^{-1}})$ | $(\mathrm{mN\,m^{-1}\,K^{-1}})$ | | $(\mathrm{mN\,m^{-1}})$ | $(\mathrm{mN\,m^{-1}})$ |
| water | 119.9 | -0.161 | Shardt and Elliott (2017) | 80.0 | 71.9 |
| oleic acid | 51.8 | -0.066 | Fig. A1 | 35.4 | 32.1 |
| glucose | 100.0 | 0 | | 100.0 | 100.0 |
| NaCl | 191.16 | -0.07188 | Janz (1980) | 173.3 | 169.7 |



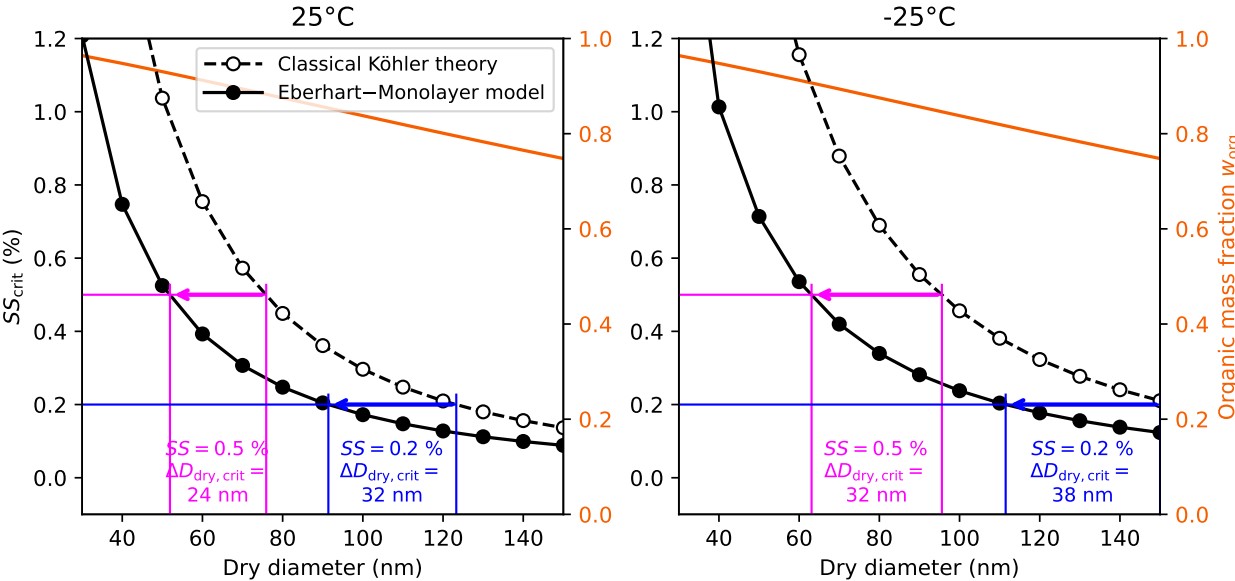

**Figure A2.** The critical supersaturation $SS_{\text{crit}}$ of SSA particles with medium organic content calculated with two different model approaches and at two different temperatures. Left: room temperature ($25\,°\text{C}$), right: $-25\,°\text{C}$. SSA particles are represented as a quaternary mixture of water, oleic acid, glucose, and NaCl. The fraction of glucose in the organic mass is $w_{\text{glu}}/w_{\text{org}} = 0.05$. The difference in the critical dry diameter $\Delta D_{\text{dry,crit}}$ between the models for a supersaturation $SS$ of $0.5\,\%$ and $0.2\,\%$ are annotated in magenta and blue, respectively.





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
