# Peer review of "The surface tension and CCN activation of sea spray aerosol particles"

_EGUsphere, 2024_

## Author Comment (AC1)

[Comment from Anonymous Referee 1] The surface tension and CCN activation of sea spray aerosol particles by Kleinheins et al. provides predictions of the critical supersaturation for sea spray aerosol. They explore a range of aerosol sizes and compositions reliant to sea spray in addition to droplets containing surfactants of varying strengths. This work uses the Eberhart-Monolayer model to describe the size- and composition-dependent surface tension during cloud droplet activation. The results are indeed interesting and agree with other model predictions. While the Eberhart model has been used to describe the surface tension of bulk solutions and the Monolayer model has been used with other surface tension parameterizations to predict the size- and composition-dependent surfactant partitioning, this work provides the first example of using them together.

[Our answers] We thank the anonymous reviewer for their comments and suggestions to improve our submitted manuscript.

However, this new combination of modeling methods does not seem to have been validated against data of size-dependent aerosol surface tension or critical supersaturations.

We have indeed performed validation calculations after having set up the model. In response to the reviewer comment, we have added now a **validation against size-dependent aerosol surface tension** using the data for surfactant-NaCl droplets by Bain et al. (2023) to the supplement. In the main text of the paper, we added the following paragraph at the end of Section 2.2:

"*To validate the presented model approach, the model was compared to surface tension measurements of 6 – 9 µm radius droplets containing a surfactant and 0.5 M NaCl, measured by Bain et al. (2023), as shown in supplement Sect. S3. Like the result by Bain et al. (2023) who used a combination of the Monolayer model with a Szyszkowski–Langmuir based surface tension isotherm, our combined Eberhart–Monolayer model reproduces the general trends well, with a tendency to overestimate bulk–surface partitioning and underestimate the surface tension.*"

In the supplement we have added the following section:

**"S3 Comparison to surface tension data of small droplets**

*Bain et al. (2023) performed surface tension measurements of 6 – 9 µm radius droplets containing one or two surfactants and a co-solute, i.e., NaCl or glutaric acid. In the same study, they combined the Monolayer model to describe partitioning with a Szyszkowski-Langmuir type equation for the surface tension isotherm to predict the surface tension of the small droplets. Similarly, here we compare our model, which uses the multi-component Eberhart model instead of the Szyszkowski-Langmuir type model, with their measurements. The results are shown in Fig. S3. Densities and molar masses were taken from Table S6 in the Supporting Information of Bain et al. (2023). The pure component surface tension of the surfactant $\sigma_2$ was chosen to be equal to the surface tension of the binary water–surfactant solution at concentrations higher than the CMC. The binary separation factor of the surfactant in water $S_{12}$ was determined by fitting the binary Eberhart model to binary water–surfactant surface tension data measured by Bain et al. (2023). Salting-out factors $A_{23}^{SO}$ and $B_{23}^{SO}$ were chosen such that the ternary Eberhart model (black lines) matches the experimental bulk data by Bain et al. (2023) (black circles). These model parameters are given at the right side of each panel. Blue solid and dashed lines show the surface tension of 6 µm and 9 µm radius droplets predicted by the Eberhart–Monolayer model, respectively. Like the results by Bain et al. (2023), our model seems to predict a slightly too strong surface partitioning leading to lower surface tensions for small droplets in most cases, but the general trends are reproduced well.*

[Figure]

*Figure S3: Comparison of the Eberhart–Monolayer model (lines) to surface tension measurements of droplets with a radius of 6 − 9 μm (blue circles) and large droplets ("Bulk", black circles) from Bain et al. (2023). The droplets contain 0.5 M NaCl and the surfactant annotated in the upper right corner of each panel. At the right side of each panel, the parameters used in the model for the respective surfactant are given."*

To better explain why a **validation against critical supersaturation data** from literature has not been included in our study, we added the following text at the end of the section "Uncertainties in the modelling approach":

*"As an additional validation step, future work should be directed at comparing $SS_{crit}$ predicted with the combined model to measurements of $SS_{crit}$ of lab-generated surfactant containing aerosol particles. A comparison to data from literature was not included in this study for two reasons. First, such literature data is very limited, as can be seen from the study by Lin et al. (2018), where the experimental data was not sufficient to draw a conclusion about which of their two models was more accurate. Second, in previous studies, the exact composition of the aerosol particles was not confirmed by a measurement, but taken as the composition of the solution filled into the atomizer. We suggest that a verification of the particle composition after atomization by e.g. an aerosol mass spectrometer is urgently needed for a reliable comparison to modelled $SS_{crit}$*

*values. To our knowledge, no study has yet investigated potential composition changes when surfactant-containing particles are produced with atomizers."*

Additionally, while the online AIOMFAC model is widely used to predict the water activity of aqueous aerosol, no evidence has been provided to show its utility for the strong surfactants that are modeled in this work.

See below under point #4.

While the results in this manuscript are interesting, some of the statements regarding the results may be too strong if these questions cannot be addressed. A list of specific comments are included below.

1) First paragraph of section 2.1 – the first statement is too strong. Whether the surface tension is lowered depends on the size and composition of the particle as bulk depletion can bring the surface tension back up to that of pure water, even when quantities of surfactant present are sufficient to reduce the surface tension of bulk solutions. The second statement is also subject to the depletion strength.

We thank the reviewer for their comment and revised the paragraph as follows:

*An atmospheric aerosol particle containing surfactants  can experience a lower surface tension than a particle without surfactants, and the extent of surface tension lowering depends on  the size and dilution of the particle. At low relative humidity, a  particle in a deliquesced state with a high concentration of surfactants is expected to have a low surface tension. However, with increasing humidity, the particle dilutes and as a result, the surface tension increases and approaches the value of pure water, as illustrated in detail by Davies et al. (2019). Therefore, a surface tension model is required to quantify surface tension as a function of the solution composition.*

2) Page 4 line 105 – how is the surface activity defined here?

Since "surface activity" is ambiguous, we changed the sentence to:

*Following the Eberhart model, we can fully characterize the  surface tension behaviour of substances in a binary solution with water with only two parameters, i.e., $S_{1i}$ and $\sigma_i$.*

3) Page 6 line 155 – Bain et al., 2023 provide an example of the Monolayer Model using a Szyszkowski–Langmuir type isotherm applied to quaternary aerosol droplets.

We thank the reviewer for pointing out this piece of literature. We adjusted the sentence to:

*Here, we use the multi-component Eberhart model (Kleinheins et al., 2024) in contrast to Malila and Prisle (2018) and Bain et al. 2023, who used a Szyszkowski–Langmuir based equation, which is limited to ternary and quaternary solutions of specific systems.*

4) Page 6 line 180 – what evidence is there that this simple substitution of SDS with dodecanoic acid is a reasonable approximation? Furthermore, dodecanoic acid is not one of the molecules AIOMFAC is trained with, and it has a much longer hydrophobic tail than anything in the predefined list. Is there evidence that AIOMFAC predictions for something of this size and hydrophobicity are accurate? I have similar questions about oleic acid, which also has a much longer hydrophobic tail than anything in the predefined list.

We think that the reviewer refers here to the predefined list available on the AIOMFAC webpage. This list just includes the atmospherically most relevant molecules. AIOMFAC is based on UNIFAC for organic-water mixtures, which has been trained on many more substances than the ones in this predefined list. But we agree that AIOMFAC has not been tested for its accuracy to describe aqueous surfactant solutions and it might be inaccurate for these. Furthermore, we represent SDS with dodecanoic acid, which is a non-dissociating species, while SDS can dissociate in aqueous solution, which was not discussed so far in the manuscript.

Note that the critical supersaturation is only influenced by solution non-ideality for particles that have undergone little hygroscopic growth until water saturation is reached, e.g., small particles with a very high organic share. While for classical Köhler theory, solution non-ideality and dissociation play an important role, for the model approaches with surface tension lowering the exact representation of oleic acid and SDS and the dissociation of SDS play a negligible role in almost all cases.

To clarify these points, we made the following adjustments (a-f):

a) At the end of Section 2.3 "water activity and solution non-ideality" we added the following text:

*"[... More details are given in supplement Sect. S4.] For the particles considered in this study, it is found that in most cases solution ideality can be assumed for the calculation of $SS_{\text{crit}}$. Only at a high content of amphiphilic or hydrophobic organic substances and small dry diameters, LLPS can lead to substantially higher $SS_{\text{crit}}$ values as further discussed in Sect. 4 below.*

*Sodium dodecyl sulfate (SDS)—a substance that is used in this study—poses some additional difficulties for the calculation of the Raoult effect. First, it can dissociate in aqueous solution, and second, its organic-sulfate group cannot be represented with the functional groups implemented in AIOMFAC. To address the second issue, we represent SDS with dodecanoic acid in AIOMFAC, which is a fatty acid with the same hydrocarbon chain length. We tested the influence of the degree of SDS dissociation and solution non-ideality when representing SDS with dodecanoic acid in supplement Sect. S5. It was found that in classical Köhler theory, solution non-ideality and the degree of dissociation of SDS play an important role for the calculation of $SS_{\text{crit}}$. However, in calculations considering surface tension lowering, in most cases, solution ideality can be assumed and the degree of SDS dissociation has negligible influence on $SS_{\text{crit}}$, except for small dry diameters and high organic content. Since we use SDS in the following as a model compound to represent general organic, surface-active compounds, e.g., fatty acids, which do not dissociate, we proceed with representing SDS with dodecanoic acid and also use a van't Hoff factor $v_{H,SDS} = 1$ in classical Köhler theory unless stated otherwise."*

b) In the supplement we added the following figure to the section "Influence of solution non-ideality", which shows the typical case where solution ideality can be assumed and adjusted the text as follows:

[Figure]

"*Figure S4: Four different ways of calculating the Raoult effect (see legend) in Köhler curves using three different model approaches (columns): with the Eberhart–Monolayer model, assuming no bulk depletion, and with Classical Köhler theory. Calculations are for an SDS-NaCl particle with $D_{dry} = 50\ nm$ with an organic fraction "med" ($w_{glu}/w_{org} = 0$). First row: Köhler curve and critical supersaturation (circle). Second row: water activity $a_w$ (Raoult effect) and saturation ratio of the Kelvin effect, calculated with the exponential function in Eq. 1. Third row: droplet surface tension. Fourth row: bulk composition (first column) and total composition (second and third column). The y-axis range was limited to 0–0.02 for a better visibility of the solute share. SDS is represented with dodecanoic acid in AIOMFAC and assumed not to dissociate in Classical Köhler theory.*"

*[...]*

*The result of this comparison for an SDS-NaCl particle of $D_{\mathrm{dry}} = 50$ nm particle with medium organic content is shown in Fig. S4. In the second row of all three columns, it can be seen that $a_w$ predicted with AIOMFAC-1ph (blue dashed line) is higher than the one assuming a hydrophobic surfactant (yellow dashed line) at low wet diameters. Therefore, the particle is assumed to undergo LLPS in that range and the best estimate (black solid line) follows the "surfactant hydrophobic" calculation. As soon as AIOMFAC-1ph predicts a lower $a_w$ than "surfactant hydrophobic" (blue and yellow lines are crossing), the droplet is assumed to be one homogeneous phase, and the best estimate follows AIOMFAC-1ph. This results in a local maximum in $a_w$ which also shows up as a global maximum in the Köhler curve when using Classical Köhler theory (first row, right column) and marks $SS_{crit}$ in that case (black circle). In the calculation using classical Köhler theory, the higher surface tension ($\sigma = \sigma_1$, see third row) leads to a higher Kelvin effect and a higher $SS_{crit}$ than those resulting from the first two model approaches. Yet, in all three model approaches, $SS_{crit}$ calculated considering non-ideality (black circles) is very similar to that assuming solution ideality (gray circles). This applies to most cases analyzed in this study.*

*Solution non-ideality only had a considerable influence on $SS_{crit}$ when the dry diameter is small ($D_{\mathrm{dry}} = 50$ nm) and the organic fraction is high and has a low O:C ratio. Two such examples are shown in Fig. S5 and Fig. S6 for an SDS--glucose--NaCl particle ($w_{glu}/w_{org} = 0.05$) and a pinonic acid--NaCl particle, respectively, both having a high organic fraction and $D_{\mathrm{dry}} = 50$ nm. These two examples show that in cases with high organic fraction and small dry diameters, the particles undergo little hygroscopic growth and, as a result, are still in a phase-separated state at activation leading to increased $SS_{crit}$ values compared to a calculation assuming ideality."*

c) We also added a section "**Influence of SDS dissociation**" to the supplement, which reads as follows:

*"In past studies, sodium dodecyl sulfate (SDS) often was assumed to fully dissociate in solution (e.g., Sorjamaa et al., 2004; Prisle et al., 2010). In fact, SDS can undergo dissociation in aqueous solution, the degree of which depends on the degree of dilution in water, as well as on the relative ratio of NaCl and SDS (Matijevic and Pethica, 1958). This raises the question of which van't Hoff factor should be used in calculations assuming solution ideality. Furthermore, SDS cannot be represented with the functional groups implemented in AIOMFAC, due to the organic-sulfate group, raising the question of how to implement solution non-ideality. To test the sensitivity of the critical supersaturation on the van't Hoff factor of SDS as well as its representation in AIOMFAC for solution non-ideality for the calculation of $a_w$, we tested three cases. First, we represented SDS by dodecanoic acid in AIOMFAC, which is a fatty acid with the same hydrocarbon chain length and an amphiphilic, non-dissociating molecule. Second, we assumed solution ideality with a van't Hoff factor of $v_{H,SDS} = 1$. Third, we assumed full dissociation of SDS by using $v_{H,SDS} = 2$. $SS_{crit}$ for these cases and all three model approaches is shown in Fig. S7.*

[Figure]

*Figure S7: Influence of solution non-ideality and SDS dissociation on $SS_{crit}$ of SDS–NaCl particles calculated with the three different model approaches (classical Köhler theory, the Eberhart–Monolayer model, and assuming no bulk depletion). In cases labelled "non-ideal", $a_w$ is calculated using AIOMFAC (best estimate, see Sect. S4), with SDS being represented with dodecanoic acid (non-dissociating). Cases labelled "ideal" assume $\gamma_w = 1$, full dissociation of NaCl and either no dissociation of SDS ("vH = 1") or full dissociation of SDS ("vH = 2").*

*When using classical Köhler theory, a strong influence of solution non-ideality and the degree of dissociation of SDS on $SS_{crit}$ is found, except for particles with $D_{dry} = 100\ nm$ and low organic content ($\Delta SS_{crit}$< 0.012 %). In contrast, the influence of solution non-ideality and SDS dissociation on $SS_{crit}$ is negligible in calculations with the Eberhart–Monolayer model or assuming no bulk depletion. Only at high SDS content (e.g., 98.8 % in dry mass), the critical supersaturation is much higher when representing SDS with dodecanoic acid due to LLPS. The reason of the higher influence of dissociation and non-ideality in classical Köhler theory is that all SDS is assumed to remain in the bulk thereby contributing to the Raoult effect. Furthermore, the lower surface tension in the other two model approaches leads to a stronger particle growth as a function of RH, such that at activation the Raoult effect is very close to 1 and less sensitive to the SDS representation in the model.*

d) We furthermore changed Table 2 as follows:

**Table 2.** Physical properties and surface tension parameters used in this study: density $\rho_i$, molar mass $M_i$, molar volume $v_i N_A$, van't Hoff factor $v_{H,i}$, pure component surface tension $\sigma_i$, and separation factor in water $S_{1i}$.

| $i$ | Substance | $\rho_i$ $\mathrm{kg\,m^{-3}}$ | $M_i$ $\mathrm{kg\,mol^{-1}}$ | $v_i N_A$ $\mathrm{cm^3\,mol^{-1}}$ | $v_{H,i}$ | $\sigma_i$ $\mathrm{mN\,m^{-1}}$ | $S_{1i}$ |
|---|---|---|---|---|---|---|---|
| 1 | water | 997 | 0.018 | 18.05 | 1 | 72.0 | 1 |
| 2 | propionic acid | 988 | 0.074 | 74.90 | 1 | 26.2 | 55.3 |
| 2 | glutaric acid | 1219 | 0.132 | 108.29 | 1 | 53.1 | 67.3 |
| 2 | valeric acid | 934 | 0.102 | 109.21 | 1 | 26.7 | 974.6 |
| 2 | pinonic acid | 965 | 0.184 | 190.67 | 1 | 52.1 | $3.7 \times 10^3$ |
| 2 | SDS | 1030 | 0.288 | 279.61 | 1 | 29.0 | $1.4 \times 10^4$ |
| 2 | oleic acid | 888 | 0.282 | 317.57 | 1 | 32.2 | $9.9 \times 10^6$ |
| 3 | glucose | 1301 | 0.180 | 138.36 | 1 | 100.0 | 1.81 |
| 4 | NaCl | 2090 | 0.058 | 27.75 | 2 | 169.7 | 0.848 |

e) In Section 4.2, we added the case "Classical Köhler theory, $v_H = 2$" in cyan to Figure 5 and the following sentence in the text:
*Even with full dissociation of SDS (van't Hoff factor =2, cyan symbols and line), $SS_{crit}$ increases with $w_{org}$ due to the higher molar volume of SDS.*

[Figure]

*Figure 5: [...] The curves in cyan are calculations with Classical Köhler theory assuming full dissociation of SDS, i.e., $v_{H,SDS} = 2$ for comparison to the black curves with $v_{H,SDS} = 1$.*

f) Lastly, in Section 5.1 "Uncertainties in the modelling approach", we added the sentence:
*AIOMFAC has not been specifically trained for amphiphilic organic molecules like SDS and oleic acid; however, as discussed in Sect. 2.3, the exact representation of the organic molecule in AIOMFAC has a negligible influence on the prediction by the Eberhart-Monolayer model except for the prediction of LLPS at high organic content and small dry diameters.*

5)  Is there any evidence that the Eberhart-Monolayer model accurately predicts size-dependent partitioning in aerosol? By comparison to literature droplet surface tension data for example? The quality of the underlying expression for surface tension can play a large role in the size- and concentration-dependent predictions using the Monolayer partitioning model.

We have addressed this question above in response to the general comment on this topic, where we describe the **validation against size-dependent aerosol surface tension** that we have performed.

6)  page 7 line 185  - *The model predicts* LLPS. Is there any experimental evidence that this is the case?

There is experimental evidence in the literature that "When substances with a low oxygen-to-carbon (O:C) ratio are mixed with inorganic salts in the same particle", liquid-liquid phase separation can occur (e.g. Song et al. 2012). Since SDS, pinonic acid, and oleic acid are all substances with a low O:C ratio, LLPS is expected to occur depending on composition. However, the exact onset composition of LLPS predicted by the model in our study might be subject to some uncertainty.

7)  page 13 line 210 – SDS is a solid at room temperature, does this mean that this is the only solute that a solid phase density was used? Yes.
How does using solid phase density impact the results?

We modified the end of Section 3.3:

*For SDS, the value of the pure solid specified by the manufacturer Merck KGaA has been used (Merck, 2023). Since the density of SDS in aqueous solution could deviate from the solid phase density, we tested the influence of a 10 % lower density on the results as shown in supplement Sect. S9. It was found that the influence on $SS_{crit}$ is small.*

...and added a small sensitivity calculation in the supplement (Sect. S9):

[Figure]

Figure S12: Influence of the SDS density on the results in Fig. 5. Black and gray data is calculated using $\rho_{SDS} = 1030 \ g/L$ and green data is calculated using a 10 % lower density, i.e., $\rho_{SDS} = 927 \ g/L$.

8) Page 13 line 316 – can the authors provide some context as to why 50 nm dry diameter was selected?

In Figure 4, a 50 nm dry diameter was selected as an example to illustrate the main trends. Results for other diameters are shown extensively in the following sections and in additional figures in the supplement. To emphasize the illustrative purpose of the figure, we changed the paragraph as:

*In Fig. 4, Köhler curves based on three different model setups are shown for a $D_{dry}$ = 50 nm SSA particle. The main assumptions in the three model approaches are summarized in Table 3 and described in the following using Fig. 4 as an illustrative example of the main trends.*

9) Page 20 line 444 – "solutes in an aerosol particle are best described by three properties..." can the authors clarify this statement, it is not clear to me how this was determined to be the best. What was it compared to?

We thank the reviewer for their comment. We changed the sentence to:

*Second, surface-active solutes in an aerosol particle can be characterized by three key properties determining their influence on the critical supersaturation: the binary separation factor in water $S_{1i}$, the pure component surface tension $\sigma_i$ and the molar volume $v_i N_A$.*

10) Page 20 line 446 – Can the authors quantify what SS$_{crit}$ lowering to a moderate degree and substantially compared to classical Kohler theory are?

We changed the paragraph to:

*Based on the $SS_{crit}$ results for 50 nm surfactant–NaCl particles at medium organic content in Fig. 7 ($w_{org} = 0.93, w_{glu} = 0$), we suggest to categorize organic compounds into weak, intermediate, and strong surfactants with respect to CCN activation as follows. Substances with $S_{1i} < 100$ have neglibile influence on $SS_{crit}$ at any concentration and therefore should not be considered surface-active for CCN activation (weak surfactants). For these, classical Köhler theory can be applied. Substances with $S_{1i}$ between about $100$ and $10^4$ influence $SS_{crit}$ by surface tension lowering to a moderate degree ($\Delta SS_{crit} \approx 0.25$ %) compared to classical Köhler theory (intermediate surfactants). Substances with $S_{1i} > 10^4$ and low $\sigma_i$ can be considered as strongly surface-active compounds that lower $SS_{crit}$ substantially compared to classical Köhler theory ($\Delta SS_{crit} \approx 0.5$ %). This categorization is also shown by the grey lines in Fig. 3.*

11) There are a number of places where the authors refer to surface tension data but then do not cite a source, or cite either El Haber et al., 2024 (which appears to be a review that compiles surface tension data from the literature) or Kleinheins et al., 2023 (a modeling paper again using data available in the literature). References to the primary sources where the data is found must be included.

We thank the reviewer for their comment. We checked our references to surface tension data and made several adjustments as follows.

a) In Section 2.1, references to experimental surface tension data are all general references to no specific datasets. To clarify that we do not refer to any specific data, we slightly adjusted the last sentence of Section 2.1 as follows: *The salting-out factors $A_{ij}^{SO}$ and $B_{ij}^{SO}$ can be obtained by fitting the model (Eq. 4– 6) to ternary surface tension data as shown by Kleinheins et al. (2024)."*

b) In Section 3.2, we used the data by El Haber et al. 2024 to determine $S_{1i}$ and $\sigma_i$ of the 76 organic compounds. For each compound, between one and 11 different experimental datasets have been used for the fitting, summing up to a total of 149 different sources. For simplicity's sake, we do not explicitly cite the 149 references that underlie the data for the 76 substances. However, we have now added a paragraph in the supplement's former Section S4 (now S5) that describes in more detail how the parameters $S_{1i}$ and $\sigma_i$ were determined and which data has been used:

*"The parameters $\sigma_i$ and $S_{1i}$ were determined for 76 organic substances based on the data compiled by El Haber et al. (2024) as follows. The pure component surface tension $\sigma_i$ for each substance was calculated as an average of the experimental pure component surface tensions given in Tables 1-3 in El Haber et al. (2024). If no pure component surface tension is given by El Haber et al. (2024), $\sigma_i$ was used as a fit parameter and determined together with $S_{1i}$ by fitting the binary Eberhart model (Eq. 3) to experimental binary surface tension data provided by El Haber et al. (2024) in the supplement. For the fitting, all experimental datasets given in the supplement of El Haber et al. (2024) were considered that were also used by El Haber et al. (2024) when fitting the Sigmoid model to determine their recommended data. For example, for propionic acid, a binary Eberhart model fit was made to the combined experimental data from Alvarez et al. (1997), Granados et al. (2006), and Suarez and Romero (2011), as shown in Fig. S10 in the upper left panel."*

c) Also, we have updated the former Figure S7 (now Figure S10) in the supplement to show the references of the data underlying the model fits for the seven organic model compounds:

[Figure]

d) We modified the sentence on line 297 to:

*For SDS–NaCl, $A_{SO}^{24} = 22.63$ and $B_{SO}^{24} = 2.8 \times 10^3$ were determined based on ternary solution data from Nakahara et al. (2011),  which are rather high values compared to those found for other ternary solutions that have been examined by Kleinheins et al. (2024).*

Typographic corrections

1) Page 16 line 363 – please check van't Hoff factor = 0. Typically, this is equal to 1 when something does not dissociate.

We thank the reviewer for spotting this typo. In the calculations, indeed, van't Hoff factor = 1 was used for non-dissociating substances. We corrected the typo in line 362:

[...] and  here is assumed not to dissociate (van't Hoff factor = 1), [...]

2) page 7 line 186 – CMC has not been defined

CMC has been defined on page 4, line 90.

3) Can the authors please check the reference list. The comma after the doi seems to be part of the link, so the links do not work when they are followed. There are also some references with duplicated text.

We thank the reviewer for their thorough reading. We double checked the references and made corrections where they were not formatted correctly.

---

## Author Comment (AC2)

[Comment from Anonymous Referee 2]

**General comment:**

Authors presented a very-well thought approach to deal with a very complex problem, the CCN activation of surfactant-enriched aerosol particles. The model is presented in a very fluent and clear way, easy to understand for possible users. The sensitivity analysis is performed carefully to assess dependencies to particle size and composition. However, the discussion in section 5 **could benefit from a comparison of model outputs to laboratory measurements of critical supersaturation for similar particle-systems** (e.g. [8-11]).

Even when I understand that this task is challenging due to the scarce data along the size-composition range of atmospheric relevant systems, the statements related to underestimation of CCN activity of ultrafine SSA particles in climate models are too strong without a proper validation of model results. Even if particles with 50 nm-diameter activate at 0.5% supersaturation level in updrafts, they could also deactivate in downdrafts leading to non significant changes in cloud droplet number concentrations at cloud base.

Nonetheless, I recommend the manuscript for publication after addressing the comments due to the completeness of the modelling approach.

Its future implementation in cloud models could bring valuable information about the formation of marine stratocumulus in pristine areas where sea spray emissions from leads can be richer in organic compared to those from open oceans. If statements in this study become proven, it would be necessary to reformulate how SSA emissions are depicted in the marine boundary layer. Even if statements do not hold, being nonactivated, surfactant-enriched SSA could be transported vertically promoting the formation of mixed-phase clouds at higher altitudes in pristine atmospheres. This is particularly important to improve our understanding of the Artic amplification phenomenon.

[Our answers] We thank the anonymous reviewer for their comments and careful reading.

To better explain why a **validation against critical supersaturation data** from literature has not been included in our study, we added the following text at the end of the section "Uncertainties in the modelling approach":

*"As an additional validation step, future work should be directed at comparing $SS_{crit}$ predicted with the combined model to measurements of $SS_{crit}$ of lab-generated surfactant containing aerosol particles. A comparison to data from literature was not included in this study for two reasons. First, such literature data is very limited, as can be seen from the study by Lin et al. (2018), where the experimental data was not sufficient to draw a conclusion about which of their two models was more accurate. Second, in previous studies, the exact composition of the aerosol particles was not confirmed by a measurement, but taken as the composition of the solution filled into the atomizer. We suggest that a verification of the particle composition after atomization by e.g. an aerosol mass spectrometer is urgently needed for a reliable comparison to modelled $SS_{crit}$ values. To our knowledge, no study has yet investigated potential composition changes when surfactant-containing particles are produced with atomizers."*

**Minor comments :**

1. Line 86 : It is important to include here more information about the pure component surface tension required to perform calculations with the model. This parameter is crucial for the model implementation. Although this is explained in detail in Kleinheins et al. (2024), the model user would benefit from a short summary of the different assumptions related to this variable. It could be useful to explore correlations based on a hypothetical supercooled liquid state for substances that are solid at atmospheric temperatures (e.g.[1-3])

We agree that the pure component surface tension is an important parameter in both the Eberhart and the Monolayer model. However, a general discussion of the choice of pure component surface tension values is beyond the scope of this study. The Eberhart model has been analyzed previously in Kleinheins et al. 2023 and 2024 showing a good performance. Also, the Monolayer model has been presented and analyzed elsewhere. Here, the focus is the application of the models to sea spray aerosol particles and less so a discussion of the physical properties of organic substances in general.

2. Line 142 and 305: The molecular volume of each substance is also crucial in the estimation of the monolayer thickness. As before, the model user would benefit from a short summary of the possible data sources and assumptions related to this variable, especially in the case of surfactants (e.g. [4])

With respect to estimating the monolayer thickness, we think that the exact value of the molecular volume is less a problem than the shape of the surfactant molecule and how they arrange at the surface. Therefore, the Monolayer model could potentially be improved by addressing these geometrical assumptions. However, this study focuses on applying the Monolayer model as is, and therefore a general discussion about the molecular volume and the assumptions behind the monolayer thickness is beyond the scope of this study. However, future work should address this topic.

3. Line 300 : Salting-out effects modify the CMC in SDS-NaCl aqueous solutions leading to minimum surface tension values below the levels observed in aqueous SDS solutions (e.g. [5-7]). This effect should be explored more if SSA model particles are going to be represented in the same way in future studies. The Eberhart model assumes that the surface tension is the linear combination of the pure compound surface tension, and even using the salting-out parameters can fail representing cloud droplet solutions along the Köhler curve, especially at low relative humidity values (e.g. RH = 95 %).

The effect of a lower surface tension at the CMC of water(1)-surfactant(2)-salt(3) solutions compared to that of water(1)-surfactant(2) solutions can indeed be captured by the Eberhart model. The higher the value of the "surface non-ideality factor for a salting-out system" $A_{23}^{\mathrm{SO}}$, the stronger the surface tension lowering at the CMC. As shown in the supplement Section "Influence of salting-out", even a value of $A_{23}^{\mathrm{SO}} = 22.63$, which is comparably high (Kleinheins et al. 2024) had no influence on the critical supersaturation. We agree that at lower RH, the salting out parameter in the surface tension term could have an effect on the hygroscopic growth of aerosol particles, yet, when the focus is on the critical supersaturation of aerosol particles, it is not relevant.

4. Figure S2 : About the mass closure. The tolerance value for the function makes use of the absolute value. I am sure that you check this, but just in case I kindly ask... have you check the sign of the individual number of moles at the droplet bulk and surface compartments?

Thanks for pointing this out. We double-checked our calculations and concluded that as long as the droplet radius is larger than one monolayer thickness $\delta$, the bulk and surface volumes give positive numbers. As a consequence, the number of moles in the bulk and at the surface are positive, too.

Would it be possible to have cases when the numerator is negative but the condition is still satisfied?

When searching numerically for a solution via iteration, the mass conservation is tried to be fulfilled as closely as possible. It is correct that in that process, both negative and positive values can appear in the numerator. However, both a negative and a positive numerator would be equally unphysical. In reality, $n_i^{bulk} + n_i^{surf}$ must be strictly equal to $n_i^{tot}$ so that the numerator is equal to zero. A small deviation from the perfect solution where mass is strictly conserved is inherent to all numerical solvers and as such not a problem of the presented solver approach.

At very small droplet sizes for pure surfactant dry particles, it is impossible to accommodate surfactant molecules in a monomolecular layer at the droplet surface compartment. The model, being just a model, tends to give negative solutions.

The limit of very small particles is surely a question of theoretical interest. Yet, such particles are too small to activate to cloud droplets and therefore not the focus of this study.

---

## Author Response (AR2)

[Comment from Anonymous Referee 1]  The authors have addressed most of my concerns in this revision. However, I still find that proper credit is not given to the authors of the surface tension data that underlies this work. If data was used in this study, the original authors deserve a citation of their paper.

[Our answers] We thank the anonymous reviewer for their second revision. We have addressed their points as described in the following.

E.g. 1 - Line 269, 306, and Fig 3 still seem to imply the underlying surface tension data is reported by El Harber et al 2024, without giving credit to the authors who actually collected and reported the underlying data.

In order to create Fig. 3, a total number of 149 different sources were used. We think that it is more convenient to cite the review where all this data has been compiled instead of citing all these papers individually. To better emphasize that El Haber et al. 2024 did not do the measurements but just reviewed the data, we adjusted the sentences as follows:

> Besides its functional groups, the surfactant model compound should be representative in its surface-active behaviour. As described in Sect. 2.1, surfactants can be characterized by their separation factor in water $S_{1i}$ and their pure component surface
> 265  tension $\sigma_i$. By fitting the binary Eberhart model (Eq. 3) to experimental surface tension data  compiled by El Haber et al. (2024), $S_{1i}$ and $\sigma_i$ were determined for 76 organic substances, which are shown in Fig. 3. The fitting procedure is described in supplement Sect. S6 and the data underlying Fig. 3 are provided in tabular form in supplement Sect. S6 and as a csv file (see code and data availability). In addition to these 76 organic compounds, $S_{1i}$ and $\sigma_i$ of atmospheric samples taken at five different locations were considered. Ekström et al. (2010) and Gérard et al. (2016) measured surface tension isotherms

> The pure component surface tension $\sigma_i$ and the binary separation factor in water $S_{1i}$ of all substances used in this study are summarized in Table 2 (see supplement Sect. S8 for underlying experimental data and model fits). For water, $\sigma_1 = 72.0\,\mathrm{mN\,m^{-1}}$ was used throughout the study. For the six model substances (propionic acid, glutaric acid, valeric acid, pinonic
> 300  acid, SDS, and oleic acid) used in the surfactant category (2), $\sigma_2$ and $S_{12}$ are taken from the fits underlying Fig. 3. The pure liquid surface tension of glucose at room temperature is not known, since glucose crystallizes at this temperature. The binary aqueous solution data from Aumann et al. (2010), Lee and Hildemann (2013), and Romero and Albis (2010) compiled by El Haber et al. (2024) is only available in a narrow concentration range. An extrapolation to supersaturated concentrations with the Eberhart model yields a very high value ($\sigma_3 > 10000\,\mathrm{mN\,m^{-1}}$) with a large uncertainty (90 % confidence interval:

We also adjusted the caption of Figure 3 as follows:

> **Figure 3.** Separation factor in water $S_{1i}$ and pure component surface tension $\sigma_i$ of organic substances (stars and 1–70: based on data  compiled by El Haber et al. (2024)) and of atmospheric samples taken at 5 different locations (71–74: Ekström et al. (2010), a–k: Gérard et al. (2016), see also supplement Sect. S6 – S8). Substances with black stars as markers are used as model compounds in this study. $S_{1i}$ was determined by fitting the binary Eberhart model (Eq. 3) to experimental surface tension data. If $\sigma_i$ was not reported in El Haber et al. (2024), it was considered an additional fitting parameter. For the atmospheric samples, $\sigma_i$ was taken as the lowest measured value. Uncertainty bars show the 95 % confidence intervals of the fit parameters. Substance names and categories are the same as in El Haber et al. (2024), i.e., the category "alcohols" also contains ketones and aldehydes. Grey lines and labels of approximate regions of weak, intermediate, and strong surfactants show the suggested categorization following the results of this study.

E.g. 2 - Fig S10 now includes references to the original datasets where the underlying surface tension data was found, but those references are not included in the reference list in the SI. If

the dataset is used in this work, the manuscript from which it was taken needs to be properly referenced in the bibliography. A reader of this manuscript should not have to go to a different paper to find the references for the data shown in the figures.

We thank the reviewer for pointing that out. We have now added these references to the reference list in the SI.

Other additional small comments:
1. line 132 - a kinetic partitioning model has also recently been validated against microscopic data (https://doi.org/10.1021/acsearthspacechem.4c00199)

We thank the reviewer for pointing to this very recent study. Knowing about it, we can no longer write that the Monolayer model is the only model validated with experimental surface tension data, and so we have adjusted the paragraph as:

In large liquid volumes, the bulk composition $x_i^{\text{bulk}}$ can be assumed to be equal to the total composition $x_i^{\text{tot}}$ (i.e., $x_i = x_i^{\text{bulk}} \approx x_i^{\text{tot}}$) in surface tension isotherms (e.g., Eberhart model). Small droplets, however, have a large surface-to-volume ratio and therefore the partitioning of substances to the surface of the droplet can lead to their depletion in the droplet bulk. To take

130   this effect into account, a partitioning model is required that introduces mass conservation and allows to quantify the bulk depletion based on physical and geometrical assumptions.  Here, we choose the Monolayer model (Malila and Prisle, 2018)  for modelling bulk–surface partitioning, which has been validated with microscopic surface tension data  in several studies (Bzdek et al., 2020; Bain et al., 2023, 2024).

2. check 'Monolayer model' is consistently capitalized

We thank the reviewer for spotting this. We have double-checked the spelling of 'Monolayer model'.